# Growth-factor-mediated coupling between lineage size and cell fate choice underlies robustness of mammalian development

Néstor Saiz[1]*, Laura Mora-Bitria[2], Shahadat Rahman[1], Hannah George[1], Jeremy P Herder[1], Jordi Garcia-Ojalvo[2]*, Anna-Katerina Hadjantonakis[1]*

[1]Developmental Biology Program, Sloan Kettering Institute, Memorial Sloan Kettering Cancer Center, New York, United States; [2]Department of Experimental and Health Sciences, Universitat Pompeu Fabra, Barcelona Biomedical Research Park, Barcelona, Spain

**Abstract** Precise control and maintenance of population size is fundamental for organismal development and homeostasis. The three cell types of the mammalian blastocyst are generated in precise proportions over a short time, suggesting a mechanism to ensure a reproducible outcome. We developed a minimal mathematical model demonstrating growth factor signaling is sufficient to guarantee this robustness and which anticipates an embryo's response to perturbations in lineage composition. Addition of lineage-restricted cells both in vivo and in silico, causes a shift of the fate of progenitors away from the supernumerary cell type, while eliminating cells using laser ablation biases the specification of progenitors toward the targeted cell type. Finally, FGF4 couples fate decisions to lineage composition through changes in local growth factor concentration, providing a basis for the regulative abilities of the early mammalian embryo whereby fate decisions are coordinated at the population level to robustly generate tissues in the right proportions.

**\*For correspondence:**
nestorsaiz@icloud.com (NS);
jordi.g.ojalvo@upf.edu (JG-O);
hadj@mskcc.org (A-KH)

**Competing interests:** The authors declare that no competing interests exist.

## Introduction

Across metazoa, coordination between cell fate specification and population size ensures robust developmental outcomes. Integration of cell behavior at the population level allows a coordinated response to injury in both embryos and adults (*Chen et al., 2015*; *Wojcinski et al., 2017*; *Young et al., 2019*). The preimplantation mammalian embryo is a paradigm of self-organization, where patterning and morphogenesis occur without the need for maternal determinants or external cues. It therefore provides an in vivo platform to understand the processes that ensure precision and robustness during the development of multicellular organisms. These embryos can tolerate cell loss, exemplified by preimplantation genetic diagnose (*Harper and Sengupta, 2012*), and can incorporate foreign cells to generate chimeric animals (*Bradley et al., 1984*; *Gardner, 1968*; *Mintz, 1964*; *Mintz and Illmensee, 1975*; *Tachibana et al., 2012*; *Tarkowski, 1961*; *Tarkowski, 1959*). Remarkably, neither of these perturbations impair embryonic development. This evidence suggests that there are mechanisms that coordinate patterning and population size to enable adaptation. Despite recent interest in understanding and exploiting the capacity of early mammalian embryos and cells for self-organization (*Bedzhov and Zernicka-Goetz, 2014*; *Deglincerti et al., 2016*; *Harrison et al., 2017*; *Morgani et al., 2018a*; *Rivron et al., 2018*; *Shahbazi et al., 2019*; *Sozen et al., 2018*; *Warmflash et al., 2014*), little is known about the local control mechanisms that enable such robust autonomous development.

The blastocyst-stage embryo is a hallmark of mammalian preimplantation development. It comprises three cell types – the pluripotent epiblast, which gives rise to the fetus, and the extra-embryonic trophectoderm (TE) and primitive endoderm (PrE, or hypoblast), which predominantly form supporting tissues (*Gardner and Rossant, 1979*; *Kwon et al., 2008*; *Nowotschin et al., 2019*; *Papaioannou, 1982*; *Viotti et al., 2014*). In the mouse, these lineages are specified during the 48 hr between embryonic day (E) 2.5 and E4.5, the time of implantation. Epiblast and PrE cells arise from a population of bipotent progenitors and comprise the inner cell mass (ICM) of the blastocyst (*Chazaud et al., 2006*; *Plusa et al., 2008*). In the mouse, epiblast specification is driven by the transcription factors NANOG, SOX2 and OCT4 (*Avilion et al., 2003*; *Chambers et al., 2003*; *Mitsui et al., 2003*; *Nichols et al., 1998*). PrE specification is driven cell-autonomously by GATA6 (*Bessonnard et al., 2014*; *Schrode et al., 2014*), which requires activation of the mitogen-activated protein kinase (MAPK) cascade downstream of fibroblast growth factor (FGF) receptors 1 and 2, stimulated by FGF4 (*Brewer et al., 2015*; *Chazaud et al., 2006*; *Frankenberg et al., 2011*; *Kang et al., 2017*; *Kang et al., 2013*; *Krawchuk et al., 2013*; *Meng et al., 2018*; *Molotkov et al., 2017*; *Nichols et al., 2009*; *Yamanaka et al., 2010*).

For development to proceed, the three cell types in the blastocyst must be specified in appropriate numbers and within a limited window of time (48 hr in the mouse, 72–96 hr in humans). In the mouse embryo, uncommitted ICM progenitors, which co-express NANOG and GATA6, adopt epiblast or PrE identity asynchronously and irreversibly over the course of blastocyst development (*Nichols et al., 2009*; *Plusa et al., 2008*; *Saiz et al., 2016b*; *Schrode et al., 2014*; *Xenopoulos et al., 2015*). In wild-type embryos, epiblast and PrE are generated in precise proportions, irrespective of the absolute size of the embryo or the ICM (*Saiz et al., 2016b*). By contrast, loss of key regulators such as *Nanog*, *Gata6*, *Fgf4* or *Fgfr1* alter these proportions and cause peri-implantation lethality (*Bessonnard et al., 2014*; *Brewer et al., 2015*; *Frankenberg et al., 2011*; *Kang et al., 2017*; *Kang et al., 2013*; *Krawchuk et al., 2013*; *Messerschmidt and Kemler, 2010*; *Mitsui et al., 2003*; *Molotkov et al., 2017*; *Schrode et al., 2014*; *Silva et al., 2009*). The ratio of these lineages is likely critical for development of the embryo beyond implantation, and therefore ICM composition must be precisely regulated (*Saiz et al., 2016b*). However, the details of such a tissue size control mechanism remain unclear.

In this study, we combine manipulations of ICM composition with predictions from in silico simulations to address the question of regulation of the number of cells allocated to each ICM lineage. We develop a minimal mathematical model in which cell fate decisions in the ICM are mediated solely by intercellular signaling. In this model, ICM cells spontaneously and robustly segregate into two lineages, which scale with embryo size as they do in vivo. The model has only two free parameters, which are adjusted to recapitulate the observed wild-type behavior. The robustness of this in silico decision is evidenced by the response of the system to perturbations that alter lineage composition. Specifically, the model predicts (with no additional parameter fitting) that reducing or increasing the number of cells in one lineage, would change the pattern of progenitor differentiation to restore lineage composition. This effect is also observed experimentally by using two-photon laser excitation for ablation of specific cells in embryos, and by adding exogenous, lineage-restricted cells to embryos to generate chimeras. The ability to recover from these perturbations is reduced over time, as the number of uncommitted progenitor cells is depleted. Finally, we alter the size of the PrE by experimentally tuning the size of the epiblast compartment. Using this system, we show that FGF4 is the growth factor providing the feedback necessary to couple lineage size with cell fate decisions. Our results provide a mechanistic basis for the regulative and scaling abilities of the early mouse embryo and illustrate how a self-organizing system can develop robustly and reproducibly without the need for external inputs.

## Results

### Cell fate decisions in the inner cell mass of the blastocyst are made at the population level

Epiblast and PrE cells originate from a population of bipotent progenitor cells that co-express the lineage-associated transcription factors NANOG (epiblast) and GATA6 (PrE) (*Chazaud et al., 2006*; *Plusa et al., 2008*; *Saiz et al., 2016b*), which we refer to as double positive (DP) cells (*Figure 1—*

*figure supplement 1A–C*). These markers can be used to automatically classify blastocyst cell types and quantify population size (*Saiz et al., 2016b*; *Saiz et al., 2016a*). PrE cells identified this way express later markers, such as SOX17 and GATA4, in a pattern consistent with previous observations (*Figure 1—figure supplement 1C–E*, *Artus et al., 2011*; *Kurimoto et al., 2006*; *Niakan et al., 2010*; *Nowotschin et al., 2019*; *Plusa et al., 2008*).

Epiblast and PrE size (with respect to cell number) scale with embryo size to maintain a consistent ICM composition (*Saiz et al., 2016b*). To determine whether this scaling is the result of an active control mechanism or a probabilistic process, we designed a biological probability test in which we mixed labeled (GFP+) with unlabeled cells (GFP-) from 8-cell stage embryos to generate series of chimeric embryos (*Figure 1A*; *Figure 1—figure supplement 2A*). Using this system, we can alter the fate of one of the two populations and assess the behavior of the other. If the epiblast and PrE lineage decisions are independent events (i.e., cell-autonomous), the differentiation pattern of the progeny of either cell population (GFP+ or GFP-) should be unaffected by the pattern of the other one.

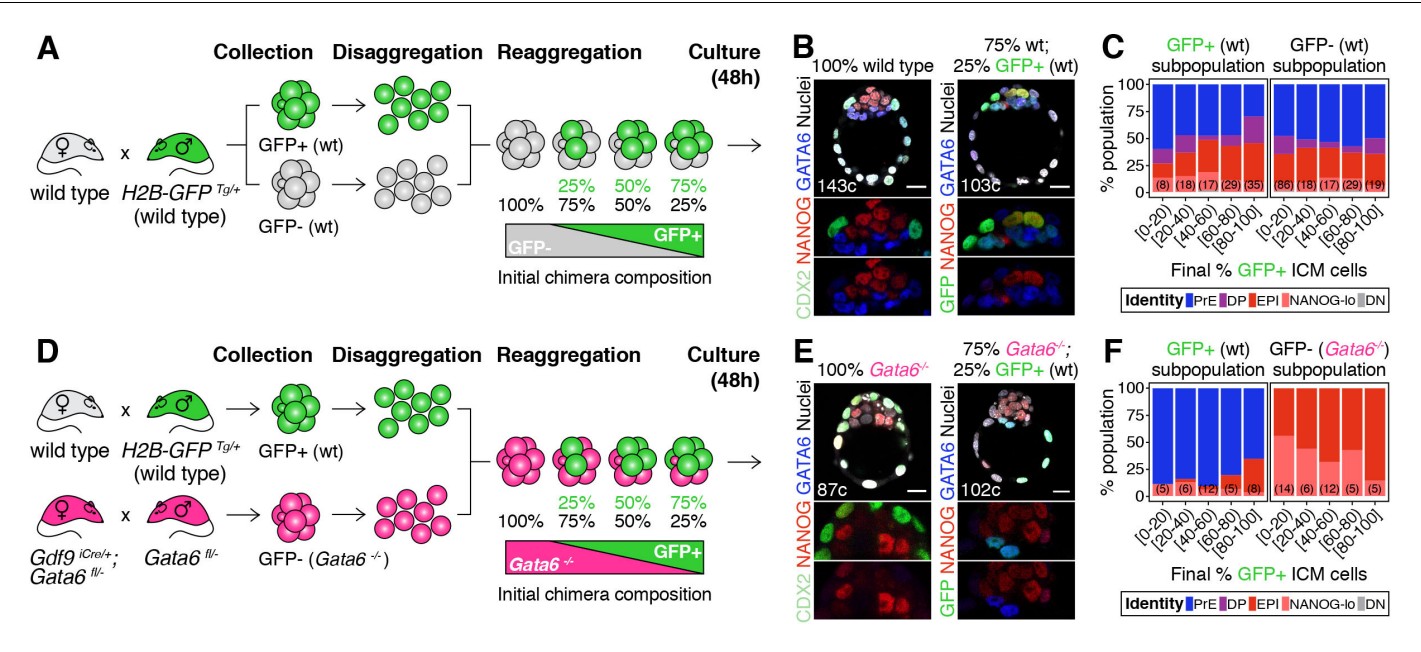

**Figure 1.** Cell fate decisions in the ICM of the blastocyst are made at the population level. (**A**) Experimental design to test independence of cell fate decisions in the ICM. Wild type embryos (H2B-GFP+ or GFP-) were collected at the 8-cell stage, disaggregated into single cells or clumps, re-aggregated in the combinations indicated and allowed to develop in culture for 48 hr, until the late blastocyst stage (equivalent to ~4.5 days post-fertilization). For further experimental details and information on alleles, see Materials and methods. (**B**) Optical cross-sections through representative immunofluorescence images of a non-chimeric, wild-type control (100% wt) and a chimera made with 75% GFP-; 25% GFP+ cells (all wild type). Magnifications of the ICM are shown below for each marker. (**C**) Stacked bar plots showing the lineage distribution of GFP+ (left) and GFP- (right) wild-type ICM cells in embryos, stratified by the final % of H2B-GFP+ ICM cells, as indicated on the x-axis. Number of embryos in each group is indicated in parentheses. (**D**) Experimental design to test independence of cell fate decisions in the ICM. Wild-type embryos (H2B-GFP+) and *Gata6$^{-/-}$* embryos were collected at the 8-cell stage, disaggregated into single cells or clumps, re-aggregated in the combinations indicated and allowed to develop in culture for 48 hr, until the late blastocyst stage (equivalent to ~4.5 days post-fertilization). For further experimental details and information on alleles, see Materials and methods. (**E**) Optical cross-sections through representative immunofluorescence images of a non-chimeric, *Gata6$^{-/-}$* control (100% *Gata6$^{-/-}$*) and a chimera made with 75% *Gata6$^{-/-}$*; 25% GFP+ (wt) cells. Magnifications of the ICM are shown below for each marker. (**F**) Stacked bar plots showing the lineage distribution of GFP+ (wt, left) and GFP- *Gata6$^{-/-}$* (right) ICM cells in embryos, stratified by the final % of H2B-GFP+ ICM cells, as indicated on the x-axis. Number of embryos in each group is indicated in parentheses. Color coding is indicated. All embryos labeled for NANOG (red), GATA6 (blue) and either CDX2 (controls) or GFP (chimeras) (green). All optical cross-sections are 5 µm maximum intensity projections. Total cell counts are indicated for each embryo within the merged images. PrE: Primitive Endoderm, DP: Double Positive (for NANOG and GATA6), EPI: Epiblast, NANOG-lo: low NANOG epiblast, DN: Double Negative (for NANOG and GATA6). Scale bars = 20 µm.

The online version of this article includes the following figure supplement(s) for figure 1:

**Figure supplement 1.** Cell type classification in the ICM of the mouse blastocyst.
**Figure supplement 2.** Cell fate decisions in the ICM of the blastocyst are made at the population level.

By contrast, if they are not, the probability of any cell adopting a particular fate will be conditional on the fate choice of other cells within the population (i.e., a non-cell autonomous decision).

We first mixed labeled and unlabeled wild type-(wt) cells, which have equivalent developmental potential, in different proportions (*Figure 1A*; *Figure 1—figure supplement 2A*). The progeny of each population contributed proportionally to TE and ICM (*Figure 1—figure supplement 2B,C*) and to all ICM cell types (*Figure 1B,C*; *Figure 1—figure supplement 2D–F*), irrespective of their representation in the resulting chimera. Chimeras made by aggregating two intact 8 —cell stage embryos (2x size) or two half embryos (1x size) showed equivalent distributions of cells (*Figure 1—figure supplement 2A–E*), further indicating that lineage allocation is independent of absolute embryo size. Next, we fixed the probability of differentiation of the GFP- population by using $Gata6^{-/-}$ cells, instead of wt cells (*Figure 1D*), and monitored the differentiation pattern of the GFP+ population (wt). $Gata6^{-/-}$ embryos are cell-autonomously unable to specify PrE cells, and exclusively form TE and epiblast (*Figure 1E*, left) (*Bessonnard et al., 2014*; *Schrode et al., 2014*). When combined with wt GFP+ cells (*Figure 1D*), $Gata6^{-/-}$ cells give rise to morphologically normal chimeras (*Figure 1E*, right; *Figure 1—figure supplement 2G–I*). In these chimeras, $Gata6^{-/-}$ ICM cells make only epiblast, as expected (*Figure 1F*, right; *Figure 1—figure supplement 2J,K*) however, the differentiation pattern of the wt compartment (GFP+) changes: wt cells now become biased towards PrE (*Figure 1F*, left; *Figure 1—figure supplement 2K*), despite having unrestricted differentiation potential. Moreover, the fate of the wt ICM cells depends on the number of $Gata6^{-/-}$ cells present: in chimeras with 40% or more mutant cells (<60% wt GFP+ cells), wt cells contribute almost exclusively to the PrE, whereas in chimeras with fewer mutant cells (>60% wt GFP+ cells), wt cells contribute to both the epiblast and PrE, and generate ICMs with a normal lineage composition (*Figure 1F*, left; *Figure 1—figure supplement 2K,L*).

Taken together, these results show that the chance of wt ICM cells adopting an epiblast or PrE fate is conditional on the fate choice made by other ICM cells (*Figure 1—figure supplement 2M,N*). Thus, lineage decisions in the ICM are not independent events, but rather are made at the population level, supporting the notion that intercellular communication coordinates cell behavior to ensure an appropriate ICM composition.

## A minimal model of cell fate decisions solely mediated by growth factor signaling explains robust lineage specification in the ICM

The experimental observations described show that cell fate choice in the ICM is non-cell autonomous. Furthermore, the lineage distribution rapidly trends to a well-defined balanced ratio (close to 50:50) of epiblast:PrE cells (*Figure 1—figure supplement 1A,B*; *Figure 2—figure supplement 1A–C*), characterized by embryo-to-embryo variability that decreases over time (*Figure 2—figure supplement 1D*). From a dynamical systems perspective, these features are indicative of the existence of a balanced attractor state within the population of proliferating cells. To test whether cell-cell signaling is sufficient to generate such an attractor, we developed a mathematical model (see *Box 1*) in which a cell fate switch is controlled by the levels of a growth factor, which itself is under the explicit control of a lineage-specific transcription factor. In the context of the blastocyst, FGF4 and NANOG, respectively, fit these criteria (*Figure 2A* and Appendix 1), although this model can be generalizable to any circuit with similar characteristics.

In our model, the state of the system is described by a single variable $x_i$ per cell, which in our case can be considered to represent the amount of NANOG in that cell. FGF4 is assumed to be activated by NANOG and feeds back onto NANOG and GATA6 via ERK (*Figure 2A* and see Appendix 1). As a result, cell-cell signaling effectively drives an indirect mutual inhibition between NANOG and GATA6, in such a way that the level of NANOG in a given cell is inhibited by that in neighboring cells (see *Box 1*), invoking a mechanism reminiscent of lateral inhibition. As in lateral inhibition, the model exhibits an attractor state in which two cell clusters coexist (*Collier et al., 1996*), corresponding to states of low and high NANOG (high and low GATA6, respectively). The simplicity of this system allows us to interpret its behavior geometrically, in analogy with earlier work on the interaction between EGF and Notch signaling (*Corson and Siggia, 2017*). The associated phase-plane portrait for our scenario is shown in *Figure 2B*, in which the two stable states are represented by black circles toward which NANOG levels tend with time (gray arrows). The situation depicted in *Figure 2B* corresponds to a perfectly symmetric case in which the epiblast:PrE ratio is 50:50 and

## Box 1. A minimal model of growth factor-mediated cell-fate decisions.

We use the following model to represent the dynamics of a population of cells in which the transcription factors NANOG and GATA6 mutually repress each other through extracellular growth factor signaling:

$$\frac{dx_i}{dt} = \frac{\alpha(1+x_i^n)^m}{(1+x_i^n)^m + \left(\frac{\langle x_i \rangle}{K}\right)^{2m}} - x_i, \quad i = 1, 2 \cdots, N$$

Here, $x_i$ represents the (dimensionless) concentration of NANOG in cell $i$, and $\langle x \rangle_i$ denotes the average value of $x$ over the immediate neighborhood of cell $i$, including $x_i$ itself. The dynamics described by the equation above correspond to motion in a potential that is double-well shaped for large enough values of the mean field $\langle x \rangle$, which is the situation of the DP fate. The two potential wells represent the high and low NANOG states, and therefore, our model implies that the DP fate is not a stable equilibrium of the cell, but a transient state towards a stable epiblast or PrE fate.

This model can be derived from a specific molecular-level circuit explicitly involving NANOG, GATA6 and the growth factor FGF4, and implicitly ERK signaling downstream of FGF receptors, as detailed in Appendix 1. However, due to its minimal character, the model is not unique to the molecular interactions assumed to be involved in this cell fate decision; other molecular circuits can likely be reduced to it to achieve the same result.

In the model, the level of NANOG is inhibited by that of its neighbors, in a manner that resembles lateral inhibition-mediated signaling. We ask whether such a model can sustain solutions in which cells cluster into two distinct cell types, expressing NANOG at two different levels (which we could interpret as epiblast and PrE cells). Epiblast and PrE cells display a salt-and-pepper distribution in the ICM at early blastocyst stages (*Chazaud et al., 2006*). Therefore, we assume perfect mixing among the cells within the population, in which case such a two-cluster state would be described by the following two-dimensional dynamical system:

$$\frac{dx_a}{dt} = \frac{\alpha(1+x_a^n)^m}{(1+x_a^n)^m + \left(\frac{x_a+x_b}{2K}\right)^{2m}} - x_a$$

$$\frac{dx_b}{dt} = \frac{\alpha(1+x_b^n)^m}{(1+x_b^n)^m + \left(\frac{x_a+x_b}{2K}\right)^{2m}} - x_b$$

The dynamics of this potential two-cluster solution can be examined via the phase-plane portrait shown in *Figure 2B*, which displays the nullclines of the system in which each of the derivatives is zero (in red and blue), the typical flow exhibited by the system from different initial conditions (in gray) and the equilibrium points of the system, two of them stable (black circles) and the third one unstable (white circle). Note that the two stable equilibria correspond to the same behavior of the system, since the labels $a$ and $b$ of the two clusters are interchangeable. The existence of the two symmetric stable equilibria ensures that the two-cluster state is a solution of the system, and that the population splits spontaneously into two distinct fates, as we show by means of agent-based simulations (described in Appendix 1) throughout the text.

cells are perfectly mixed (see *Box 1*), and as we go on to show, the attractor is robust to variations in this perfect balance.

We next sought to determine how cell fate decisions are established according to the model, when cells divide (thereby increasing in number) and are rearranged as the embryo develops and

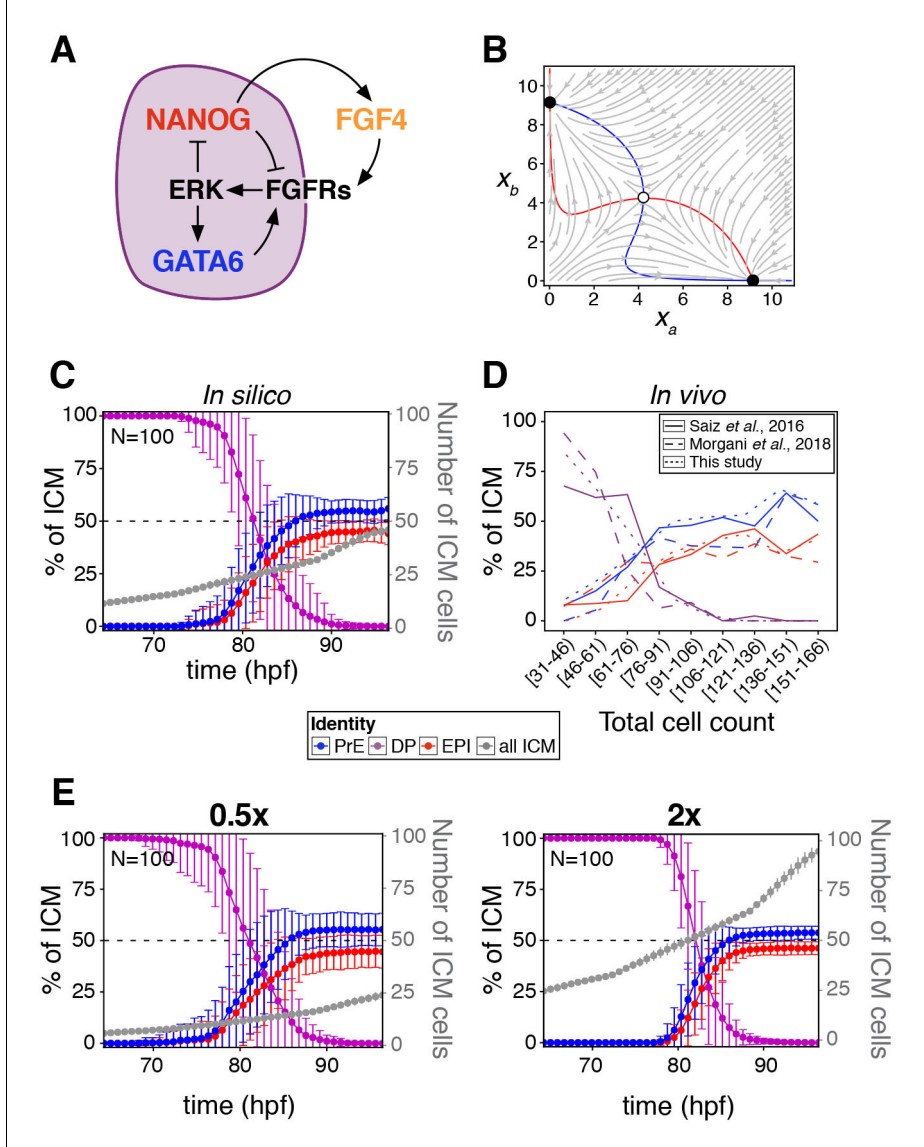

**Figure 2.** A minimal model of cell fate decisions solely mediated by growth factor signaling explains robust lineage specification in the ICM. (**A**) Diagram of our proposed model of molecular control of cell fate in the ICM. (**B**) Phase plane of our model for the case of a two-cluster state. Each axis shows the NANOG levels in one of the two clusters. Red and blue lines show the nullclines of the system (*Box 1*); the gray arrows depict typical trajectories of the system; and black (and white) circles correspond to stable (and unstable) equilibria of the two-cluster system. (**C**) Lineage dynamics in in silico simulations of ICM development using our proposed model. (**D**) In vivo ICM lineage dynamics from three experimental datasets, as indicated. (**E**) Lineage dynamics in in silico simulations of scaling experiments (to be compared with the experimental results of *Saiz et al., 2016b*). Absolute ICM size was modified to 0.5 or 2x the normal size, as indicated.

The online version of this article includes the following figure supplement(s) for figure 2:

**Figure supplement 1.** A minimal model of cell fate decisions solely mediated by growth factor signaling explains robust lineage specification in the ICM.

grows. To that end, we implemented an agent-based model to simulate the growth of the ICM (see Appendix 1) in three dimensions, in which cells divide and interact with one another via a soft-sphere potential (*Tosenberger et al., 2017*). The biochemical model described is applied to the interacting cells, and the system is allowed to progress biochemically and dynamically until a fixed cell number is reached. Simulations show that a stable cell fate distribution is consistently reached (*Figure 2C*;

frame 0

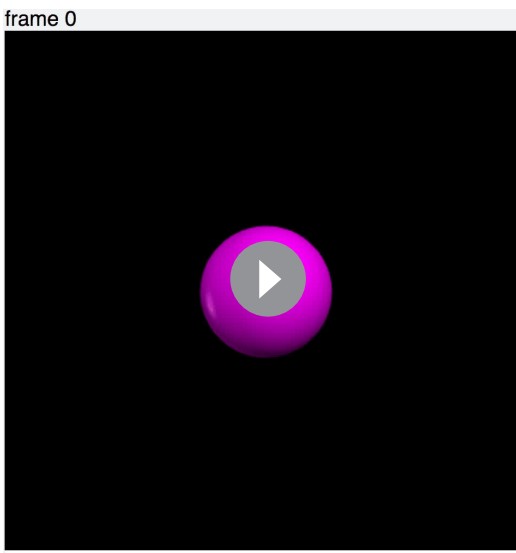

**Animation 1.** Simulation of ICM formation and lineage specification using our minimal mathematical model.
https://elifesciences.org/articles/56079#video11

*Animation 1*), in agreement with the described attractor dynamics, and in line with experimental observations from this study and others (*Figure 2D*). The response of the system is not dependent on the dimensionality of the cell population, as a two-dimensional layer of cells can also reach a balanced decision (*Figure 2—figure supplement 1E*). A sensitivity analysis of the 3D system (*Figure 2—figure supplement 1F*) shows that the behavior reported is robust to changes in the modeling parameters, both biochemical and mechanical. In particular, increasing the range of FGF signaling from nearest neighbors to a globally coupled situation in which all cells in the embryo see each other, does not qualitatively alter the outcome of fate decisions across the population (*Figure 2—figure supplement 1G*). It is worth noting that such a global coupling limit could be interpreted as a reduced Turing mechanism in which activation is local, while the inhibitor exhibits unrestricted diffusion (i.e. is widespread).

The fate decisions ascribed by the model are also biologically robust, since altering the absolute cell number within the in silico ICM leads to scaling of lineage size to maintain ICM composition (*Figure 2E*), in agreement with experimental observations (*Saiz et al., 2016b*). These results suggest that growth factor-mediated feedback is sufficient to endow the embryo with robustness to perturbations. Reaching this conclusion required us to uncouple cell-cell signaling interactions from intracellular (cell-autonomous) mechanisms, something not possible experimentally, in vivo, and which would be challenging to do in more detailed biochemical models (*Bessonnard et al., 2014*; *De Mot et al., 2016*; *Tosenberger et al., 2017*).

## The lineage composition of the ICM is robust to expansion of the epiblast

The attractor solution found in the model described suggests that the cell fate decisions reached by the embryo are robust to perturbations, in particular to those affecting the size of each population. To probe the capacity of the system to perceive and adjust to changes in cell numbers, we expanded the epiblast by introducing increasing amounts of mouse embryonic stem cells (ESCs) into embryos (*Figure 3A*). ESCs are derived from the epiblast and exclusively contribute to this lineage when re-introduced into an embryo (*Beddington and Robertson, 1989*; *Boroviak et al., 2014*; *Brook and Gardner, 1997*; *Lallemand and Brûlet, 1990*; *Nagy et al., 1990*; *Tokunaga and Tsunoda, 1992*). To control for variation in the contribution of ESCs to chimeras, we stratified embryos based on the final size of the ESC compartment, relative to the average size of the epiblast in controls (1x control EPI = 5–10 cells, 2xEPI = 10–20 cells, etc.) (*Figure 3—figure*

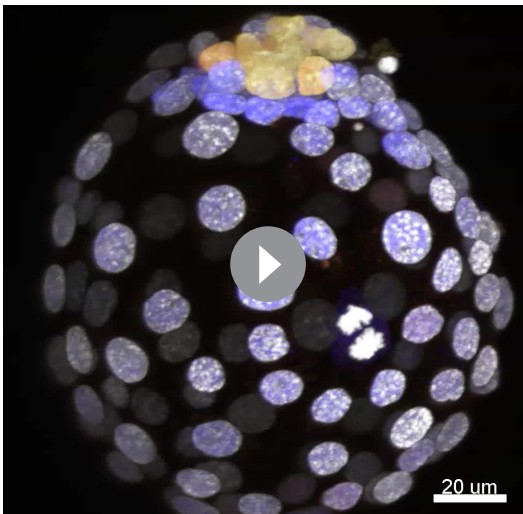

**Video 1.** Wild-type chimera with an ESC compartment equivalent to 2x control epiblast (added at morula stage). Yellow surface: ESCs, blue nuclei: GATA6, white nuclei: Hoechst.
https://elifesciences.org/articles/56079#video1

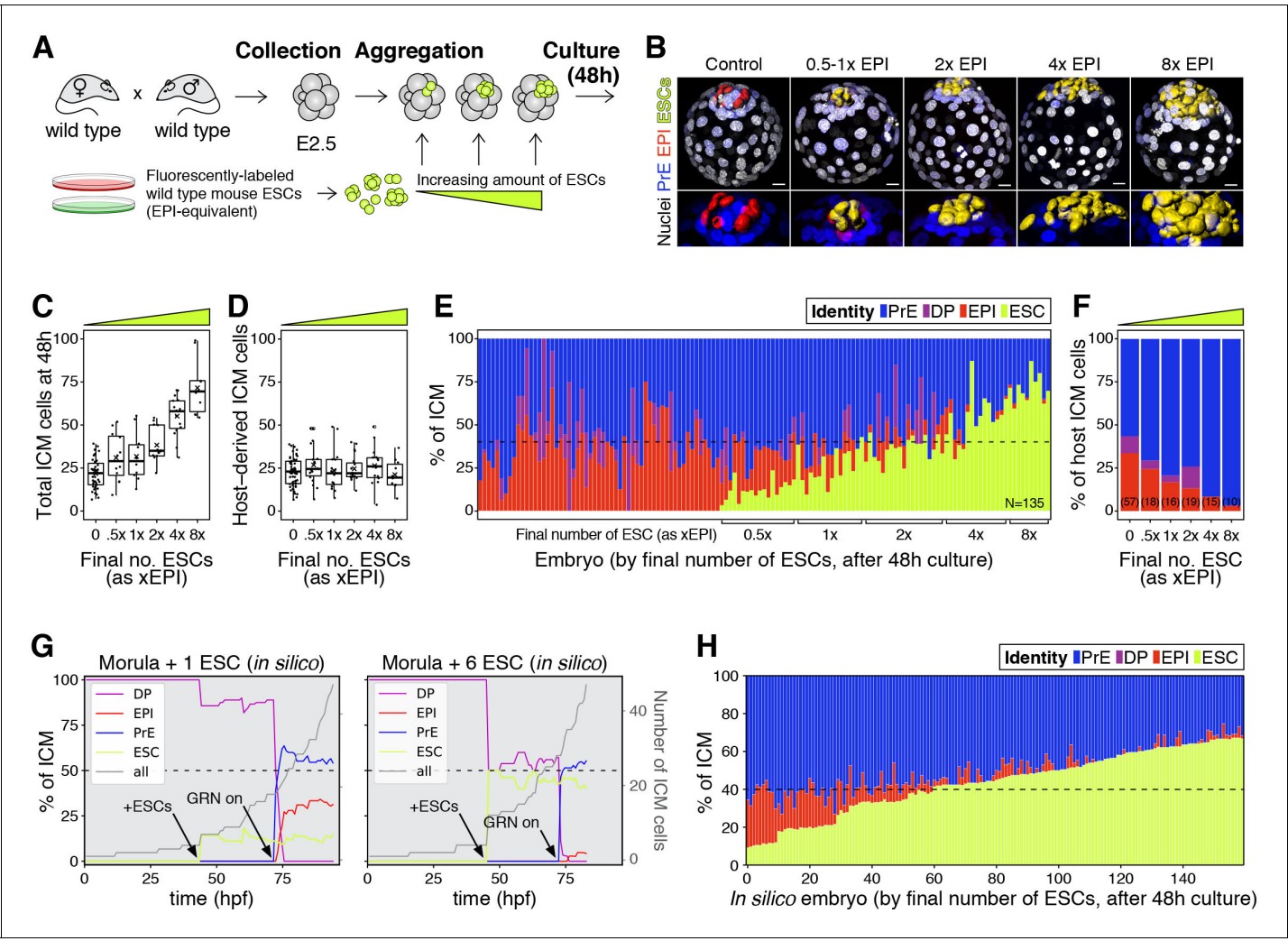

**Figure 3.** The lineage composition of the ICM is robust to expansion of the epiblast. (**A**) Experimental design. 8- or 8–16 cell stage embryos (2.5 days post-fertilization) were recovered from CD1 (wild type) females crossed with CD1 males. Embryos were denuded, aggregated with clumps of fluorescently labeled ESCs and cultured for 48–56 hr, until the late blastocyst stage (equivalent to ~4.5 days post-fertilization). (**B**) 3D renders of a series of control (no ESCs) and chimeras, generated as indicated in (**A**), carrying increasing amounts of ESCs (indicated as EPI-equivalent size (x EPI)). Control epiblast and ESCs in chimeras are highlighted as computer-rendered volumes and color coded as indicated. GATA6+ PrE is shown in blue, NANOG+ host epiblast is shown in red where applicable. (**C**) Box plot showing the total size of the ICM (host-derived + ESCs) at the end of the experiment in each group of embryos, as defined by the size of the ESC compartment. (**D**) Box plot showing the size of the host-derived ICM component at the end of the experiment in each group of embryos, as in (**C**). (**E**) Stacked bar plot showing the relative ICM composition at the end of the experiment for all embryos analyzed. Each bar represents the ICM of one embryo, ordered by increasing absolute number of ESCs at the end of the experiment. Dashed line indicates the normal ratio of 60% PrE:40% epiblast found in intact wild-type embryos. Number of embryos analyzed is indicated (N). Brackets on x-axis indicate the number of ESCs in those embryos, relative to the size of the average wt control epiblast (xEPI). (**F**) Stacked bar plot showing the relative contribution of host cells to each of the ICM lineages in each group of embryos. Yellow wedge represents the increasing amount of ESCs in each group. Number of embryos in each group is indicated in parentheses. (**G**) Growth curves showing lineage dynamics in in silico simulations of the aggregation experiments shown in (**A**). Left Y-axis and curves for each lineage indicate relative size (as % of ICM). Right Y-axis and gray curves indicate total number of ICM cells (including ESCs). (**H**) Stacked bar plot showing the relative ICM composition at the end of the experiment in in silico simulations of the experiments shown in (**A**) and (**E**). Each bar represents the ICM of one simulated embryo (i.e., a single iteration), and bars are arranged by increasing absolute number of ESCs at the end of the simulation, as in (**E**). Dashed line indicates the normal ratio of 60% PrE:40% epiblast found in intact wild-type embryos. Color coding is indicated for (**E, G, H**). In all box plots whiskers span 1.5x the inter quartile range (IQR) and open circles represent outliers (values beyond 1.5x IQR). Crosses indicate the arithmetic mean and each dot represents one embryo. Yellow wedges represent the increasing amount of ESCs in each group. PrE: Primitive Endoderm, DP: Double Positive (for NANOG and GATA6), EPI: Epiblast, ESC: embryonic stem cell. Scale bars = 20 µm.

The online version of this article includes the following figure supplement(s) for figure 3:

**Figure supplement 1.** The lineage composition of the ICM is robust to expansion of the epiblast.

*supplement 1A*). In this way, we generated chimeras with increasingly larger epiblasts and asked how ICM composition was affected (*Figure 3B*; *Figure 3—figure supplement 1B*; *Videos 1* and *2*).

Increasing the number of ESCs in chimeras increased the overall size of the ICM (host + ESCs) (*Figure 3C*), although it did not affect the contribution of the host embryo to the ICM (*Figure 3D*) or the ratio between TE and host ICM compartments (*Figure 3—figure supplement 1C*). This result suggests that ESCs have no net effect on the survival or proliferation of host cells. By contrast, increasing the number of ESCs in chimeras reduced the contribution of host cells to the epiblast (*Figure 3E–F*; *Figure 3—figure supplement 1D–E*) and increased their contribution to the PrE (*Figure 3F*; *Figure 3—figure supplement 1F*). This shift resulted in maintenance of a normal ICM composition in chimeras comprising as many ESCs as 2-4xEPI, but not more (*Figure 3E*). The ICM composition of chimeras comprising an equivalent of 4xEPI, was in most cases comparable to that of *Gata6$^{+/-}$* or *Fgf4$^{+/-}$* blastocysts, which exhibit a reduction in PrE cell numbers relative to epiblast (~40% PrE, 60% EPI; *Figure 3E*; *Figure 3—figure supplement 1D*), but which develop into viable and fertile adults (*Kang et al., 2013*; *Krawchuk et al., 2013*; *Schrode et al., 2014*). These data suggest that an expansion of up to 4x wt of the epiblast compartment is compatible with normal development.

We next asked whether this experimental perturbation would be recapitulated by our mathematical model in silico. To do so, increasing amounts of epiblast-equivalent cells (ESCs) were added to the system before activation of the molecular circuit ('GRN on', *Figure 3G*, arrows). These cells produce FGF4 and thereby impact NANOG dynamics in their neighbors but, being lineage restricted, are not subject to the same regulatory dynamics as host cells. In agreement with our experimental results, these ESCs effectively contribute to the epiblast compartment to maintain the overall ICM composition for a wide range of perturbations (*Figure 3G,H*). In particular, as the number of ESCs added is increased, the number of host cells that acquire an epiblast fate is progressively reduced, until the epiblast compartment is eventually composed only of ESCs. Notably, all these perturbations shift cell fate choice in the ICM toward PrE without net ICM growth, indicating that there is no compensatory proliferation in this context and that the regulative capacity of the system is mediated only by changes in cell fate allocation. Together, these experimental and modeling results underscore the robustness of the system and further establish a population-level coordination of cell fate choice.

## The lineage composition of the ICM is robust to in silico reduction of lineage size

The chimera experiments described allow us to probe the response of the embryo to perturbations at a fixed time, early in the process, but not as progenitors are depleted over time (*Figure 1—figure supplement 1A,B*). However, an attractor state such as the one described in *Figure 2* should also be robust to perturbations all along the system's trajectory, as long as they are not exceedingly large. Experimentally, it is not possible to expand the epiblast using ESCs at sequential stages of blastocyst development (not shown). Instead, to probe the ability of the system to adjust to perturbations over time, we used our model to modify ICM composition in silico to different degrees and at sequential time points. With that goal in mind, we first eliminated 30% of the cells of all three ICM lineages (epiblast, PrE and DP) at a time when each represents ~1/3 of the ICM. This perturbation, which is equivalent to scaling down the absolute size of the ICM by 30%, does not alter the relative composition of the ICM (*Figure 4A*). As in our simulations of embryo scaling (*Figure 2E*), the

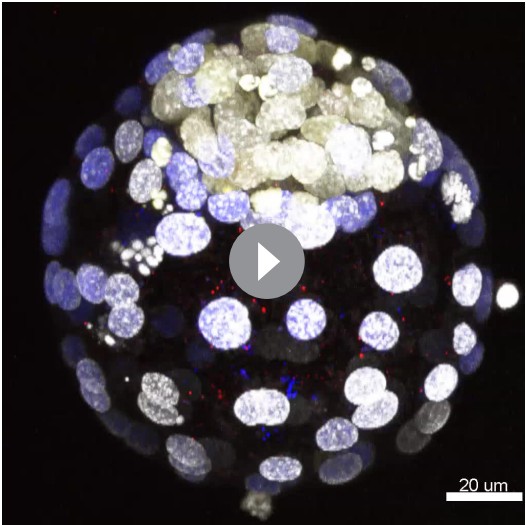

**Video 2.** Wild-type chimera with an ESC compartment equivalent to 8x control epiblast (added at morula stage). Yellow surface: ESCs, blue nuclei: GATA6, white nuclei: Hoechst.
https://elifesciences.org/articles/56079#video2

relative ICM composition at the end of the simulation was maintained robustly across embryos (*Figure 4B*). Eliminating cells in this way (equally across cell types) had no effect on ICM composition irrespective of the developmental stage at which it was carried out or the magnitude of the perturbation (*Figure 4C*).

We then eliminated PrE or epiblast-biased cells only, therefore causing an acute deviation in the normal ICM ratio. When we removed 100% of either lineage at the same developmental stage as above, we observed only a partial recovery of the targeted lineage (*Figure 4D–G*). However, a partial reduction (−30%) in either lineage was completely compensated for, and normal ICM composition was restored at all developmental stages (*Figure 4H–I*, left). By contrast, the ability to recover from loss of 100% of a lineage was reduced over time, as progenitor cells are lost (*Figure 4H–I*, right), indicating that the ability of the system to recover from changes in ICM composition depends on both the presence of uncommitted progenitor cells and the magnitude of the perturbation.

## Laser ablation enables alteration of ICM composition with high spatiotemporal control in mouse embryos

To experimentally validate the predictions of our mathematical model, we used a multi-photon laser to eliminate cells and thus alter lineage composition in live embryos (*Figure 5A,C* and see Materials and methods). Laser cell ablation is routinely used in non-mammalian systems to eliminate cells in a non-invasive way. In mammalian embryos, however, while it has been previously applied to disrupt tissues (*Eiraku et al., 2011*; *Reupke et al., 2009*; *Takaoka et al., 2017*), its potential to target single cells and its effect on developmental competence has not been determined. We combined two spectrally distinct nuclear reporters to identify all cell types in the ICM: a green fluorescent reporter for *Pdgfra* expression (*Pdgfra*$^{H2B-GFP/+}$ [*Hamilton et al., 2003*; *Plusa et al., 2008*]) and a ubiquitous red fluorescent nuclear mKate2 reporter (*Susaki et al., 2014*). PrE and uncommitted progenitors are labeled with different levels of GFP (*Plusa et al., 2008*; *Xenopoulos et al., 2015*), whereas epiblast cells are identified by the presence of nuclear mKate and the absence of GFP (*Figure 5B,D*; *Figure 5—figure supplement 1A–D*; *Videos 3–4*) (see Materials and methods for details on cell identification and classification). We verified the death of targeted cells using 3D time-lapse imaging (*Figure 5E*; *Video 5*) and estimated their half-life to be 3.4 hr (*Figure 5F*). The use of a two-photon laser allowed targeting of individual nuclei on any z-plane of the ICM without affecting other cells in the light path. Untargeted cells in both ablated and control embryos showed identical survival (*Figure 5F*), including those immediately adjacent to targeted cells (*Figure 5G*). The gross morphology and size of ablated embryos was comparable to that of intact controls, even after targeting 100% of either lineage (*Figure 5H*; *Figure 6—figure supplement 1B*).

## The cell fate choice of uncommitted ICM progenitors is dictated by lineage size

We used laser ablation (*Figure 5A,C*) to test how recovery from changes in ICM composition correlates with developmental time and magnitude of perturbation (*Figure 4F–G*). We eliminated increasing numbers of PrE or epiblast cells (2–18 cells, representing 50–100% of either lineage, *Figure 6—figure supplement 1A*) at sequential stages of blastocyst development (embryos comprising from 50 to 110 total cells) and assessed the composition of the ICM after a recovery period of 24 hr (*Figure 6A*). Although embryos showed a reduction in ICM size as a result of the perturbation (*Figure 6—figure supplement 1C*), they contained both PrE and epiblast cells (*Figure 6B*), indicating at least a partial ability to recover. Predictably, their ability to re-establish a normal PrE:epiblast ratio was reduced the later the developmental stage at the time of cell ablation (*Figure 6C–D*; *Figure 6—figure supplement 1D*). As in our simulations (*Figure 4H*), this deviation was more pronounced the larger the magnitude of the perturbation (*Figure 6C–D*; *Figure 6—figure supplement 1D*).

We next assessed the response of progenitor cells to changes in ICM composition. We tracked individual cells in time-lapse movies for 15 hr following cell ablation. Progenitor cells in intact embryos contribute to both epiblast and PrE, as assessed by *Pdgfra* expression (*Figure 5—figure supplement 1F*; *Figure 6E*). We found a bias of these cells toward PrE in intact embryos (*Figure 6E*), likely due to oversampling of PrE (given their co-expression of *Pdgfra* and mKate2), as well the overrepresentation of PrE cells within the ICM. Progenitor cells in embryos where the PrE was targeted showed a comparable trend to control embryos, generally upregulating *Pdgfra*

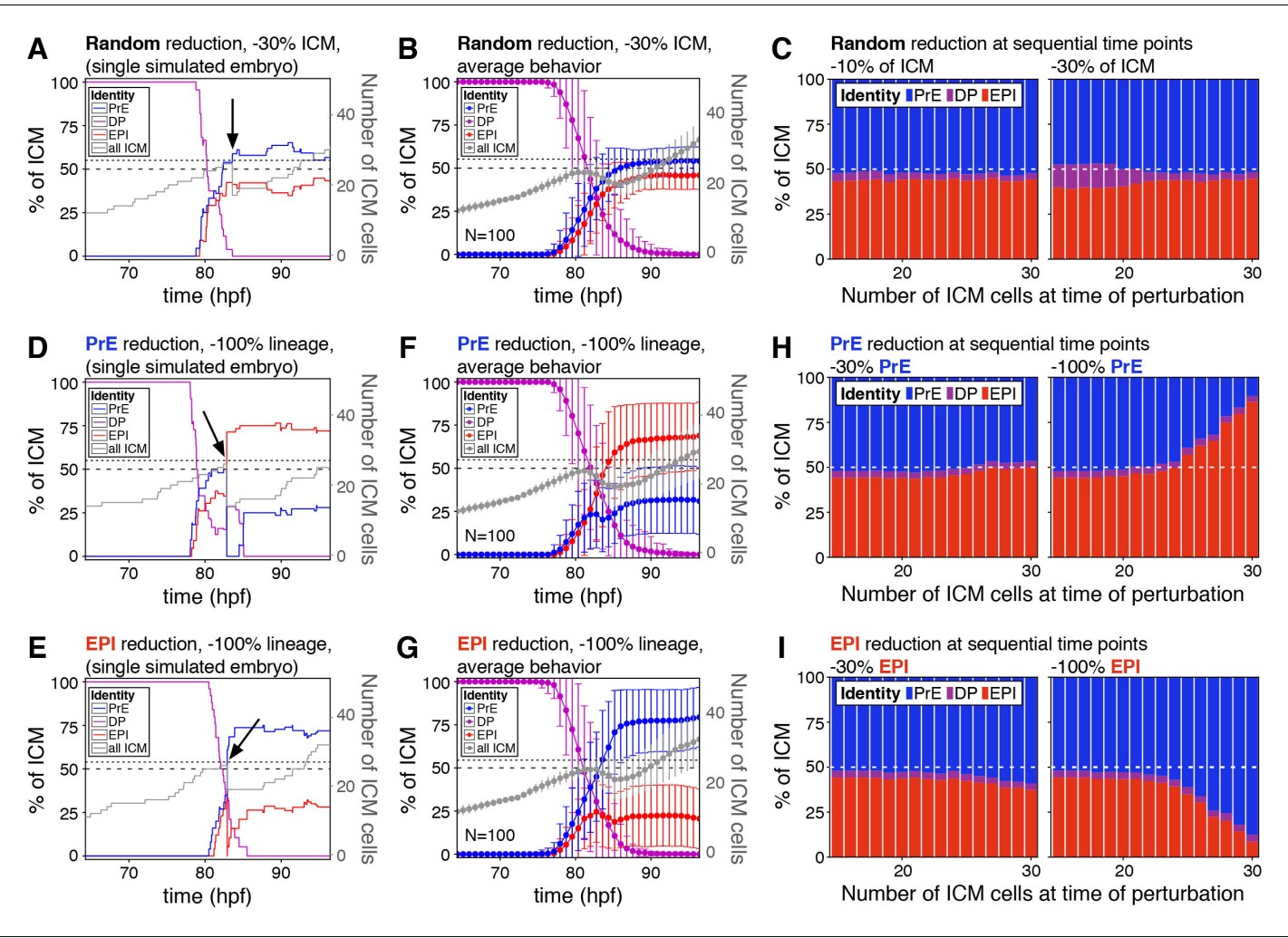

**Figure 4.** The lineage composition of the ICM is robust to in silico reduction of lineage size. (A) Growth curves for each ICM lineage after simulation of a 30% reduction in ICM size (3 PrE, 3 DP and three epiblast cells removed from a 27-cell ICM) using our model described in *Figure 2*. Arrow indicates the time of cell elimination. Lines are color-coded for each lineage, as indicated and represent relative lineage size (scale on the left Y-axis). Grey line indicates the absolute size of the ICM, as shown on the right Y-axis. Dotted line indicates 27 ICM cells, the point at which cells were eliminated. Dashed line indicates 50% of the ICM, for reference. (B) Growth curves as those in (A) showing the average behavior for 100 simulations. Error bars indicate the standard deviation. (C) Stacked bar plots showing the final ICM composition after simulating the elimination of 10% (left) or 30% (right) of ICM cells at sequential points in embryo development. Developmental stage at the time of cell elimination is indicated on the x-axis as number of ICM cells (15–30 ICM cells, equivalent to ~50–100 total cells). Each bar represents the result of 100 simulations. (D, E) Growth curves for each ICM lineage after simulation of a 100% reduction in PrE (D) or epiblast (E), when the ICM reaches 27 cells, as shown in (A) and indicated by the arrow. Lines are color-coded for each lineage, as indicated and represent relative lineage size (scale on the left Y-axis). Grey line indicates the absolute size of the ICM, as shown on the right Y-axis. Dotted line indicates 27 ICM cells. Dashed line indicates 50% of the ICM, for reference. (F, G) Growth curves as those in (D, E) showing the average behavior for 100 simulations of PrE (F) and epiblast (H) reduction. Error bars indicate the standard deviation. (H, I) Stacked bar plots showing the final ICM composition after simulating the elimination of 30% (left) or 100% (right) of the PrE (H) or the epiblast (I) at sequential points in embryo development. Developmental stage at the time of cell elimination is indicated on the x-axis as number of ICM cells (15–30 ICM cells, equivalent to ~50–100 total cells). Each bar represents the result of 100 simulations. Color coding is indicated. hpf: hours post-fertilization, PrE: Primitive Endoderm, EPI: epiblast, DP: double positive.

expression and acquiring a PrE identity (*Figure 6E*, purple line). Conversely, in embryos where epiblast cells were eliminated, progenitors had a propensity to downregulate or maintain low levels of *Pdgfra* and become epiblast (*Figure 6E*, purple line). Accordingly, at the end of movies, cells initially identified as progenitors were more frequently found in the lineage that had been targeted (*Figure 6F*), and the final PrE:epiblast ratio (in movies where we could track all or most of the ICM

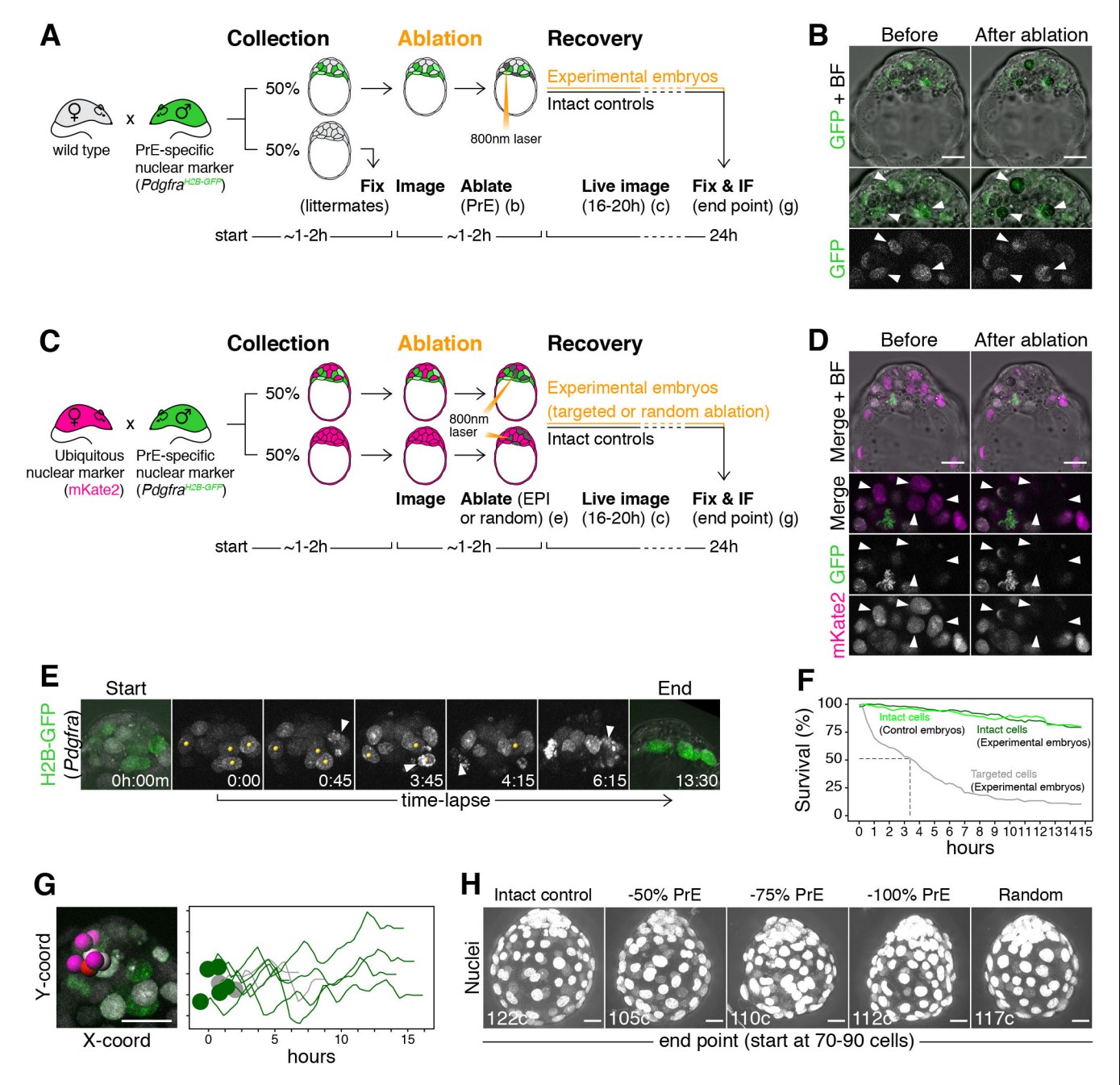

**Figure 5.** Laser ablation enables alteration of lineage size with high spatiotemporal control in mouse embryos. (**A**) Experimental design for PrE ablation. Blastocysts recovered from CD1 (wild type) females crossed with $Pdgfra^{H2B-GFP/+}$ or $Pdgfra^{H2B-GFP/+}$; $R26:CAG:3x-nls-mKate2^{Tg/Tg}$ males were sorted for GFP. GFP- embryos were fixed as reference littermates to estimate the developmental stage of the entire litter. GFP+ embryos were used for the experiment and subject to ablation of different amounts of PrE cells (GFP+) followed by 16–20 hr of live imaging to visualize the response to cell ablation, followed by 8–4 hr of in vitro culture (for a total of 24 hr). (**B**) Representative images of live GFP+ embryos before and after ablation of PrE cells. Bottom panels show magnifications of the ICM with GFP alone on grayscale, as indicated. Arrowheads point at targeted PrE cells. (**C**) Experimental design for epiblast ablation. Blastocysts were recovered from $R26:CAG:3x-nls-mKate2^{Tg/Tg}$ females crossed with $Pdgfra^{H2B-GFP/+}$; $R26:CAG:3x-nls-mKate2^{Tg/Tg}$ males. GFP-, mKate2+ embryos were used as either intact or random ablation controls. GFP+, mKate2+ embryos used for the experiment and subject to ablation of different amounts of epiblast cells (GFP-) or PrE cells (GFP+), followed by 16–20 hr of live imaging to visualize the response to cell ablation, followed by 8–4 hr of in vitro culture (for a total of 24 hr). (**D**) Representative images of live GFP+ embryos before and after ablation of epiblast cells. Bottom panels show magnifications of the ICM, for both markers together and each of them on grayscale, as indicated. Arrowheads point at targeted epiblast cells. (**E**) Still images of a representative embryo in the hours after ablation. See also *Video 3*. Yellow spots on grayscale images mark targeted cells. Arrowheads point at cell death of each targeted cell. All images are timestamped as h:mm. (**F**) Survival

*Figure 5 continued on next page*

Figure 5 continued

curves for targeted (gray) and intact cells in both experimental (dark green) and intact control embryos (light green). Dashed line marks half-life of targeted cells (~3 hr) (**G**) Survival of intact cells neighboring targeted cells. Image shows selected ICM cells (intact DP and epiblast cells, color coded, and targeted PrE cells, gray). Graph shows X-Y coordinates of cells shown in picture and survival (hours) for each cell. Time scale for each cell is shifted based on their initial X position, for visualization purposes. (**H**) Maximum intensity projections of representative embryos fixed after 24 hr in culture, showing all nuclei over bright field image. Treatment was done at the 70–90 cell stage and is indicated above. Total cell count for each embryo is shown within each image. PrE: Primitive Endoderm, EPI: Epiblast. Scale bars = 20 μm.

The online version of this article includes the following figure supplement(s) for figure 5:

**Figure supplement 1.** Lineage composition and PrE reporter dynamics in fixed and live embryos.

cells) was comparable to that obtained with end-point analysis experiments (*Figure 6G and C–D*; *Figure 6—figure supplement 1D and E*).

Finally, we investigated the relative contribution of each cell behavior (cell death, division and specification of progenitors) to recovery of ICM composition after cell ablation. In addition to changes in progenitor specification (*Figure 6E,F*), we observed both cell proliferation and cell death in the PrE and epiblast compartments for each embryo we could track (*Figure 6—figure supplement 2A–E*). Cell ablation increased the survival of progenitor cells, although it had no notable effect on either intact PrE or epiblast cells (*Figure 6—figure supplement 2F*). Conversely, we observed a slight increase in progenitor proliferation following PrE, but not epiblast ablation (*Figure 6—figure supplement 2G*). Lastly, in the sampled embryos where we targeted 100% of the PrE (N = 3), we found a high degree of cell death that was not compensated for by proliferation or progenitor specification (*Figure 6—figure supplement 2B*), and which resulted in an inability to restore ICM composition (*Figure 6E–F*).

Overall, these results indicate that changes in ICM composition are primarily compensated for by a shift in the differentiation pattern of progenitor cells. Although the small sample size precludes any definitive conclusion regarding the roles of cell death and proliferation, both our cell ablation and chimera data suggest that cell death and division play only accessory roles in this process, and that uncommitted progenitors are the primary substrate for regulation.

## FGF4 provides the dynamic readout of lineage size that determines cell fate

We have shown here that the lineage composition of the ICM is robust to changes in absolute tissue size, both in vivo and in silico, and we propose that growth factor-mediated feedback ensures this robustness (*Figure 2C*). Fibroblast growth factor 4 (FGF4), secreted by ICM cells, is necessary for PrE specification (*Kang et al., 2013*; *Krawchuk et al., 2013*), making it an attractive candidate for the feedback signal. To test this hypothesis, we used $Fgf4^{-/-}$ embryos, which cannot make PrE (*Video 6*), to generate chimeras in which we introduced increasing amounts of wt ESCs, which provide a localized source of FGF4 and are a proxy for wt epiblast cells (*Figure 7A*). Thus, increasing amounts of ESCs allow us to titer the process of PrE specification.

We found that wt ESCs can rescue PrE specification in $Fgf4^{-/-}$ embryos (*Figure 7B*; *Figure 7—figure supplement 1A*; *Video 7*). Conversely, $Fgf4^{-/-}$ ESCs produced chimeras without PrE, phenocopying $Fgf4$ null embryos (*Figure 7B*; *Figure 7—figure supplement 1A*; *Video 8*) and confirming FGF4 as the ligand responsible for the rescue. wt ESCs contributed to chimeras with $Fgf4^{-/-}$ and wt host embryos comparably (*Figure 7—figure supplement 1B*; *Figure 3—figure supplement 1A*) and, although we observed more variation in the size of the host-derived ICM compartment in wt ESC ↔ $Fgf4^{-/-}$ embryo chimeras than in wt ESC ↔ wt

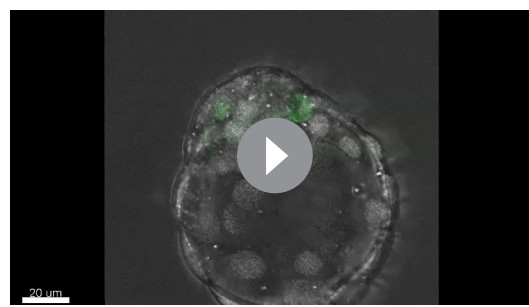

**Video 3.** Movie of experimental embryo after ablation highlighting targeted PrE cells (gray) and intact PrE cells (blue).

https://elifesciences.org/articles/56079#video3

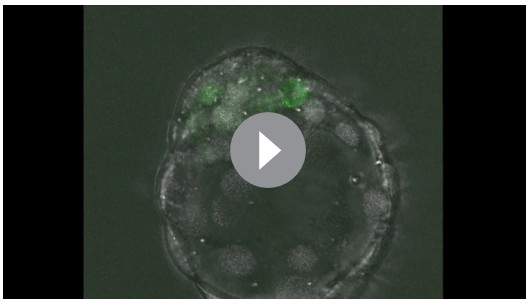

**Video 4.** Movie of experimental embryo at the 67 cell stage after ablation of 50% PrE cells, highlighting all tracked cells – PrE (yellow: targeted, blue: intact), DP (magenta), epiblast (red). Green nuclei are H2B-GFP expressed from the *Pdgfra* locus, white nuclei are nuclear mKate expressed from the ROSA26 locus (see text).

https://elifesciences.org/articles/56079#video4

**Video 5.** Movie of experimental embryo at the 97-cell stage after ablation of 50% epiblast cells, highlighting all tracked cells – PrE (blue), DP (magenta), epiblast (red). Green nuclei are H2B- GFP expressed from the *Pdgfra* locus, white nuclei are nuclear mKate expressed from the ROSA26 locus (see text). Targeted epiblast cells cannot be followed because of bleaching of nuclear mKate reporter.

https://elifesciences.org/articles/56079#video5

embryo chimeras, this did not reflect the number of ESCs present (*Figure 7—figure supplement 1C*; *Figure 3D*). However, the number of wt ESCs present in chimeras dictated the size of the PrE (*Figure 7C*). Chimeras with a number of ESCs equivalent to a wt epiblast (1xEPI) showed a full rescue of absolute PrE size (*Figure 7C*). Both the absolute and relative size of the PrE in these chimeras depended on the size of the ESC compartment (*Figure 7C–D*), with chimeras frequently exhibiting a complete rescue of ICM composition, equivalent to wt embryos (*Figure 7D*, dashed line). Of note, chimeras with a high number of ESCs (4x-8xEPI) showed a deviation from wt ICM composition comparable to that observed in wt ESC ↔ wt embryo chimeras (*Figures 7D* and *3E*; *Figure 7—figure supplement 1D,E* and *Figure 3—figure supplement 1D,E*). In both cases, ICM composition was disrupted with very high numbers of ESCs (4-8xEPI, *Figure 7D*; 3E), likely due to a lack of compensatory proliferation.

Lastly, we tested the need for progenitor cells to rescue the PrE by introducing wt ESCs into *Fgf4*⁻/⁻ blastocysts, which have totally or partially lost the DP compartment (*Figure 7—figure supplement 1G,H*, *Kang et al., 2013*; *Ohnishi et al., 2014*). Addition of ESCs at this stage failed to rescue the ICM composition in most cases, with only some chimeras specifying low numbers of PrE cells (*Figure 7—figure supplement 1H,I*; *Videos 9*, *10*). Taken together, these data demonstrate that FGF4 provides the feedback that couples lineage size and specification, acting as the cell-counting mechanism and eliciting an effect only on uncommitted progenitor cells. Such feedback control enables the system to dynamically adjust the differentiation rate of progenitors in response to perturbations in lineage specified cells to ensure a robust and consistent developmental outcome.

## Discussion

Preimplantation mammalian embryos face two major challenges: to generate sufficient numbers of cells of each of their constituent lineages and to do so before implantation into the uterus. Blastocyst lineages scale with embryo size (*Papaioannou and Ebert, 1995*; *Saiz et al., 2016b*), and perturbations in absolute cell numbers are compatible with development to term (*Mintz, 1967*; *Papaioannou et al., 1989*; *Tarkowski, 1961*; *Tarkowski, 1959*), suggesting the variable under control at preimplantation stages is relative, not absolute, lineage size. In this study, we show both theoretically and empirically that growth factor-mediated feedback couples cell fate decisions with lineage size in the mouse blastocyst to ensure consistent cell type proportions. Feedback control is widely used in complex systems to buffer noise and ensure robust behavior – in quorum sensing, it alters gene expression to coordinate cellular behaviors at the population level, from bacteria to mammals (*Balázsi et al., 2011*; *Chen et al., 2015*; *Lander, 2011*). Our data show that, in the

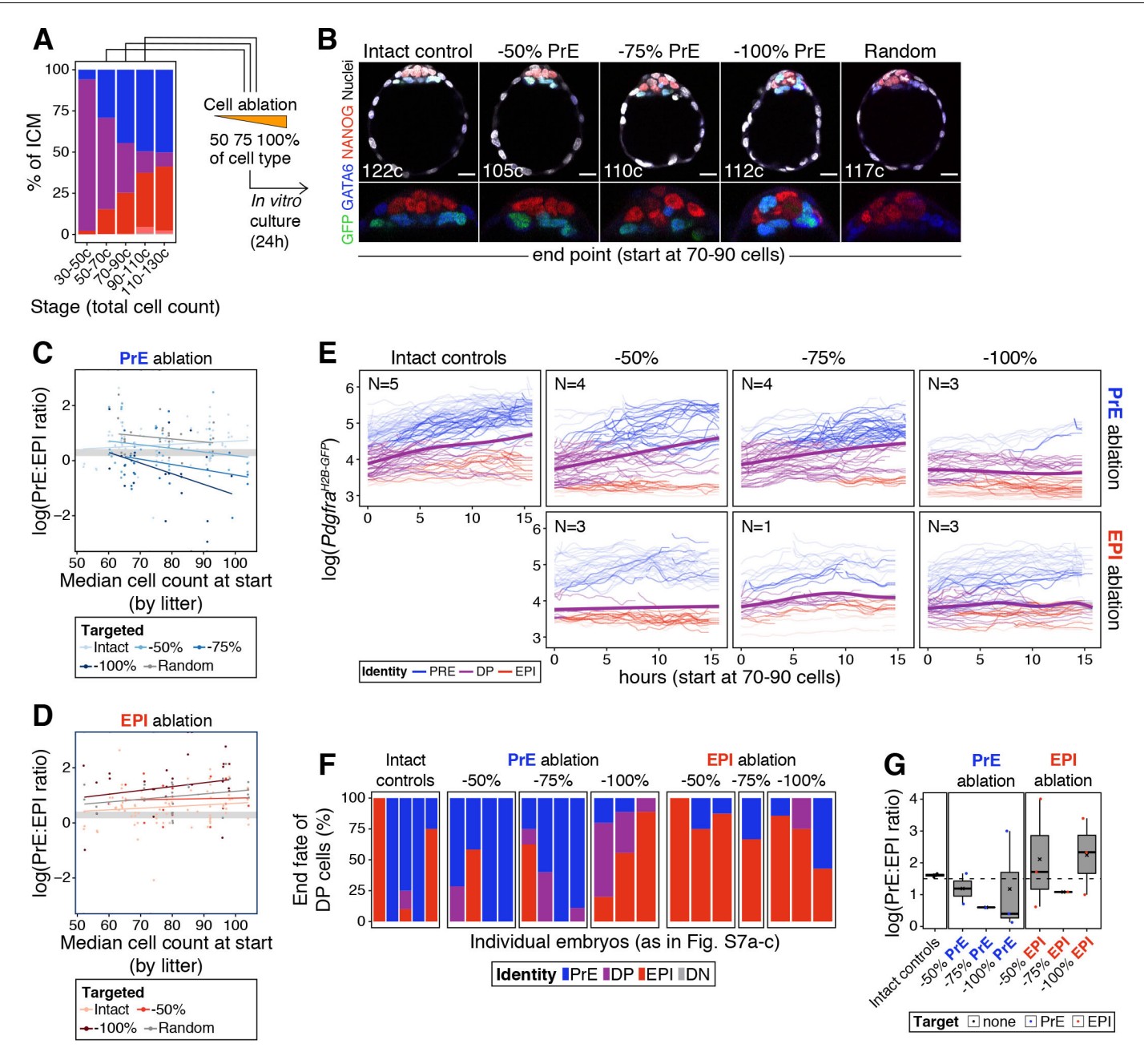

**Figure 6.** The cell fate choice of uncommitted ICM progenitors is dictated by lineage size. (A) Stacked bar plot showing ICM composition at sequential stages of blastocyst development. Embryos at each of these stages were subject to laser ablation of different fractions of either the PrE or the epiblast and allowed to recover in vitro for 24 hr (see *Figure 5A,C*). (B) Optical cross sections through representative immunofluorescence images of embryos subject to ablation at the 70–90 cell stage and fixed at the end of the experiment (24 hr later) (same embryos as in *Figure 5G*). Embryos are labeled for NANOG (red) and GATA6 (blue) to identify all ICM cell types. H2B-GFP, indicating *Pdgfra* expression, is shown in green where applicable. Lower panels show magnifications of the ICM Treatment for each embryo is indicated over each image. (C) PrE:epiblast ratio (shown as natural logarithm) at the end of the experiment (24 hr) in embryos where fractions of the PrE were eliminated at sequential stages of blastocyst development, as indicated on the x-axis. Shades of blue indicate the magnitude of the reduction in the PrE. Intact controls are embryos in which no cells were killed, Random controls are embryos in which randomly chosen ICM cells were killed without knowing their identity, in equivalent numbers to the −100% group. (D) PrE:epiblast ratio (shown as natural logarithm) at the end of the experiment (24 hr) in embryos where fractions of the epiblast were eliminated at sequential stages of blastocyst development, as indicated on the x-axis. Shades of red indicate the magnitude of the reduction in the epiblast. (E) Dynamics of *Pdgfra^{H2B-GFP}* expression in progenitor cells (DP) of experimental embryos targeted at the 70–90 cell stage. Each line represents one cell. Color coding indicates cell identity, as inferred from reporter expression (see Methods). Number of embryos per plot is indicated. Cells classified as PrE or epiblast at the beginning of the experiment are shown as color-coded semi-transparent lines behind progenitor cells, for reference. Smoothing

*Figure 6 continued on next page*

*Figure 6 continued*

curves for *Pdgfra* expression in progenitor cells are shown as thick purple lines. Fraction of the PrE or epiblast eliminated is indicated above each panel, lineage targeted is indicated to the right of each panel. (F) Stacked bar plots showing the final identity adopted by progenitor (DP) cells in each of the embryos plotted in (E). (G) Box plots showing the PrE:epiblast ratio (shown as natural logarithm) at the end of the movie, in embryos where all or most of the ICM cells could be tracked after cell ablation at the 70–90 cell stage (subset of embryos shown in (E) and (F)). Compare to panels (C) and (D). Treatment is indicated on the x-axis. In box plots whiskers span 1.5x the inter quartile range (IQR) and open circles represent outliers (values beyond 1.5x IQR). Cross indicates the arithmetic mean and each dot represents one embryo. PrE: Primitive Endoderm, DP: Double Positive (for NANOG and GATA6), EPI: Epiblast. Scale bars = 20 μm.

The online version of this article includes the following figure supplement(s) for figure 6:

**Figure supplement 1.** Population sizes and lineage composition in ablation experiments.

**Figure supplement 2.** Analysis of cell behaviors in ablation experiments.

mammalian blastocyst, growth-factor-mediated feedback ensures robust patterning independent of absolute embryo size, and it enables regeneration after injury.

Existing models of cell fate specification in the blastocyst have combined a switch mediated by transcription factors (*Huang et al., 2007*) with growth factor feedback (*Bessonnard et al., 2014*; *Nissen et al., 2017*; *Schröter et al., 2015*; *Tosenberger et al., 2017*) (reviewed in *Simon et al., 2018*; *Tosenberger et al., 2019*). However, the lack of experimental evidence for direct mutual inhibition between the transcription factors NANOG and GATA6 in the embryo, and the non-cell autonomous nature of this cell fate decision (*Figure 1*), suggest that intercellular feedback alone could drive this process. In agreement with this expectation, we show that a minimal model containing only indirect mutual inhibition via growth factor signaling is sufficient to (i) robustly generate two ICM compartments (epiblast and PrE), (ii) enable lineage scaling with embryo size, and (iii) adjust for changes in lineage composition. In our model, each cell fate promotes the differentiation of neighboring progenitors toward the opposite fate, thereby ensuring a balanced cell type composition. This behavior is consistent with the observation that embryos with defective activation of the FGF4-MAPK cascade have an excess of epiblast cells (*Brewer et al., 2015*; *Chazaud et al., 2006*; *Kang et al., 2017*; *Kang et al., 2013*; *Krawchuk et al., 2013*; *Molotkov et al., 2017*; *Nichols et al., 2009*) and vice versa (*Yamanaka et al., 2010*).

Although FGF4 signaling controls lineage size in the ICM of the mouse blastocyst, our model is agnostic to the nature of the growth factor involved and could be readily applied to binary cell fate decisions in other contexts. Notably, cell fate proportions during *Dictyostelium discoideum* development are controlled through the secreted factor DIF-1 in an analogous manner (*Kay and Thompson, 2001*), and members of the TGF-β family mediate negative feedback during skeletal muscle and olfactory epithelium specification in the mouse (*McPherron et al., 1997*; *Wu et al., 2003*). In the blastocyst of other mammalian species, NANOG and GATA6 are involved in this fate decision, but the requirement for FGF signaling is less clear (*Kuijk et al., 2012*; *Piliszek et al., 2017*; *Roode et al., 2012*; *Soszyńska et al., 2019*), suggesting a role for other signaling pathways, or a cell autonomous fate decision. Comparing the prediction of our model, others with alternative configurations, and the result of experimental perturbations, should help elucidate the relative contribution of intercellular signaling and transcription factor networks to lineage specification in these contexts.

The regulative ability of the mouse blastocyst has been extensively tested (*Gardner, 1968*;

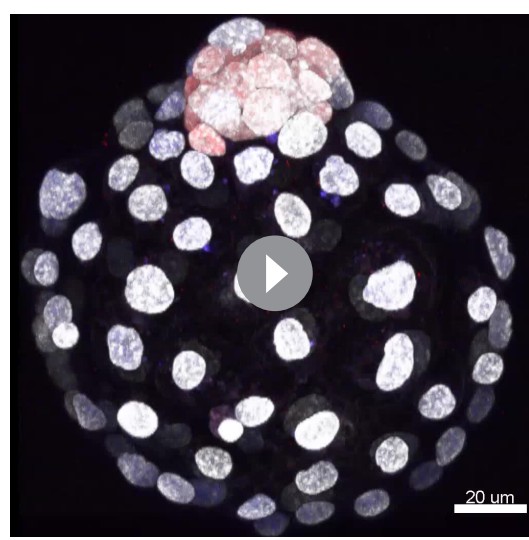

**Video 6.** *Fgf4- / -* control blastocyst. Red surface: epiblast (NANOG+), blue nuclei: GATA6, white nuclei: Hoechst.

https://elifesciences.org/articles/56079#video6

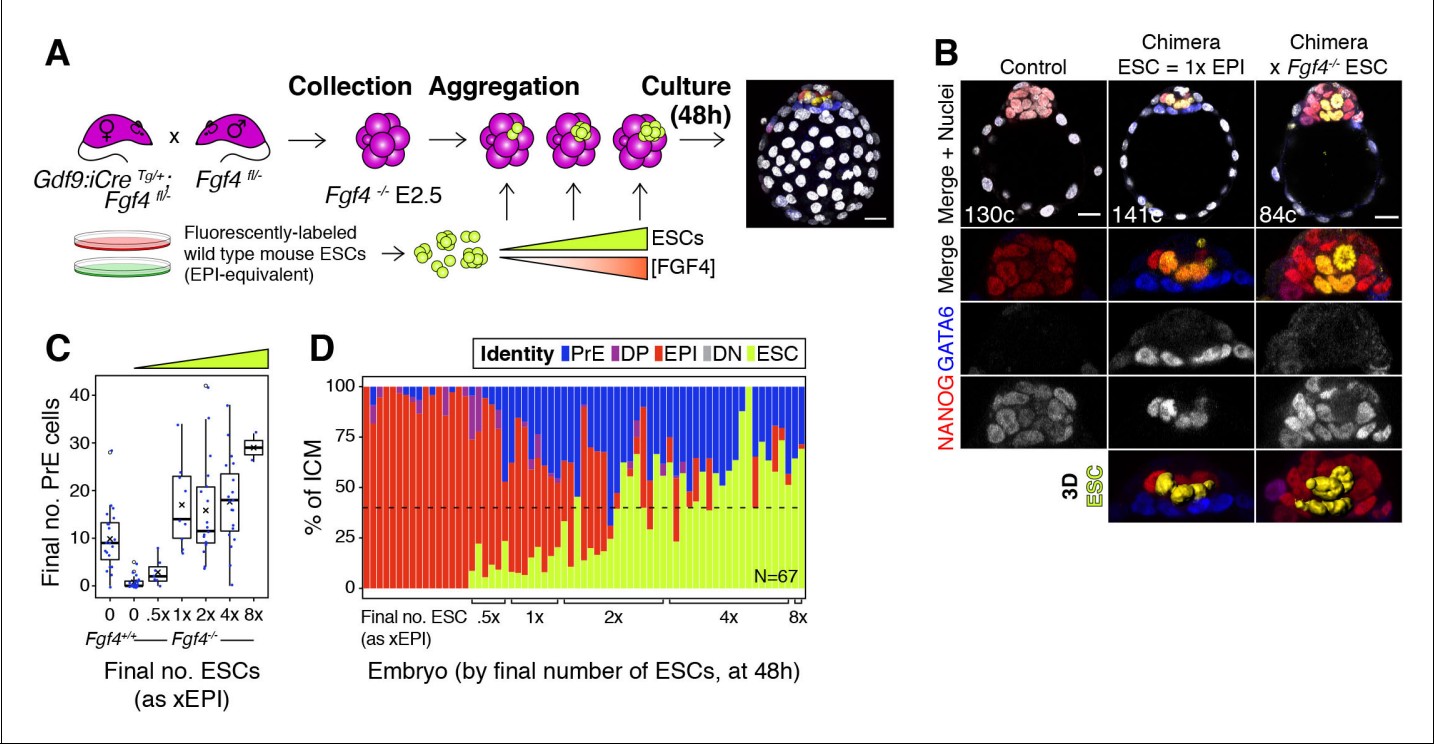

**Figure 7.** FGF4 provides the dynamic readout of lineage size that determines cell fate specification. (**A**) Experimental design. Maternal zygotic (mz) *Fgf4*$^{-/-}$ Embryos were recovered from *Gdf9*$^{iCre/+}$; *Fgf4*$^{fl/-}$ females crossed with *Fgf4*$^{fl/-}$ males at the 8cell stage (2.5 days post-fertilization). Embryos were aggregated with clumps of fluorescently labeled wild-type ESCs and cultured for 48–56 hr, until the late blastocyst stage (equivalent to ~4.5 days post-fertilization). Control embryos were allowed to develop without adding ESCs. Both chimeric and non-chimeric control embryos were fixed at the end of the culture period and labeled with lineage markers to assess ICM composition. (**B**) Optical cross-sections through representative chimeras carrying either wild type or *Fgf4*$^{-/-}$ fluorescently labeled ESCs (as indicated) and non-chimeric control embryos labeled for NANOG and GATA6 to identify all ICM cell types. The progeny of the introduced ESCs is labeled in yellow. Total cell count is indicated for each embryo. Lower panels show magnifications of the ICM, with all markers overlaid and for each individual marker in grayscale. Surface renders of ESC compartment within the ICM are shown below. (**C**) Box plots showing absolute number of PrE cells after 48 hr in wild type, control embryos (no ESCs), *Fgf4*$^{-/-}$ control embryos (no ESCs) and *Fgf4*$^{-/-}$ chimeric embryos, grouped by the size of the ESC compartment, as in *Figure 3*. (**D**) Stacked bar plot showing the relative ICM composition in individual embryos (controls or chimeras). Each bar represents the ICM of one embryo and bars are arranged by absolute number of ESCs present in the embryo. Brackets on x-axis indicate the number of ESCs in those embryos, relative to the size of the average wt control epiblast (xEPI), same groups as in (**C**). Color coding is indicated. All optical cross-sections are 5 µm maximum intensity projections. In all box plots whiskers span 1.5x the inter quartile range (IQR) and open circles represent outliers (values beyond 1.5x IQR). Cross indicates the arithmetic mean and each dot represents one embryo. Yellow wedges represent the increasing amount of ESCs in each group. PrE: Primitive Endoderm, DP: Double Positive (for NANOG and GATA6), EPI: Epiblast, DN: Double Negative (for NANOG and GATA6), ESC: embryonic stem cell. Scale bars = 20 µm.

The online version of this article includes the following figure supplement(s) for figure 7:

**Figure supplement 1.** FGF4 provides the dynamic readout of lineage size that determines cell fate specification.

*Krupa et al., 2014*; *Mintz, 1967*; *Tarkowski, 1961*; *Tarkowski, 1959*). Introduction of ESCs into morula-stage embryos is used to generate mice entirely derived from ESCs (*Lallemand and Brûlet, 1990*; *Nagy et al., 1990*; *Poueymirou et al., 2007*; *Tokunaga and Tsunoda, 1992*), which bias host cells toward extra-embryonic lineages (*Humięcka et al., 2016*). By contrast to what we observe here, introduction of ESCs also affected the host contribution to the ICM in a previous study (*Humięcka et al., 2016*). This discrepancy could be due to differences in the time points analyzed, the absence of physical constraints imposed by the zona pellucida in our experiments and/or the higher number of cells introduced by Humięcka and colleagues. We have proposed that a critical element in the regulative ability of the blastocyst-stage embryo is the asynchrony in progenitor specification toward epiblast or PrE (*Saiz et al., 2016b*). Our model reproduces this asynchronous character of cell fate allocations. This results partly (but not exclusively, *Figure 2—figure supplement 1H*) from asynchrony in cell division, which is introduced in the model by randomly perturbing the cell division time around its average value (*Tosenberger et al., 2017*). Cell cycle phase has been

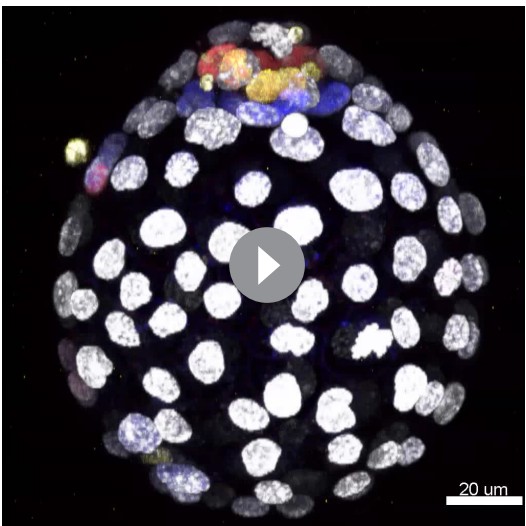

**Video 7.** *Fgf4- / -* chimera with an ESC compartment equivalent to 1x control epiblast (added at morula stage). Yellow surface: ESCs, red nuclei: NANOG, blue nuclei: GATA6, white nuclei: Hoechst.
https://elifesciences.org/articles/56079#video7

linked to fate decisions in pluripotent stem cells (*Pauklin and Vallier, 2013*), and it has been shown that heterogeneity in cell cycle phase ensures a robust cell-type composition in *Dictyostelium* (*Gruenheit et al., 2018*). In the blastocyst, asynchrony in cell cycle phase might enable a dynamic response to changes in growth factor levels, whereby only the subset of progenitors competent to differentiate at any given point will respond to the perturbation. This asynchrony in the cell cycle may ultimately underlie both the progressive allocation of cell fates observed in the ICM and the ability of the system to respond to perturbations in lineage composition.

We probed the robustness of this system by introducing lineage-restricted cells into the embryo to generate chimeras, or by decreasing lineage size using laser cell ablation. Our mathematical model exhibits attractor dynamics through which progenitor cells differentiating after the perturbation would adopt the fate needed to restore the lineage balance, a prediction corroborated by our experimental perturbations. A key element in this response is the fact that NANOG levels in a cell are inhibited by those in its neighbors, through a lateral inhibition mechanism. Such growth-factor-mediated lateral inhibition can explain the salt-and-pepper distribution of cell types in the ICM (*Chazaud et al., 2006*; *Rossant et al., 2003*), although cell movement within the ICM likely precludes the observation of the canonical, alternate distribution of fates. Besides Delta-Notch signaling, lateral inhibition can also result from mechanical cues (*Xia et al., 2019*) and from secreted signaling molecules (*Thompson et al., 2004*). It has been proposed that high local concentrations of FGF4 may underlie this stochastic distribution of fates in the ICM of the blastocyst (*Bessonnard et al., 2014*; *Kang et al., 2017*; *Kang et al., 2013*;

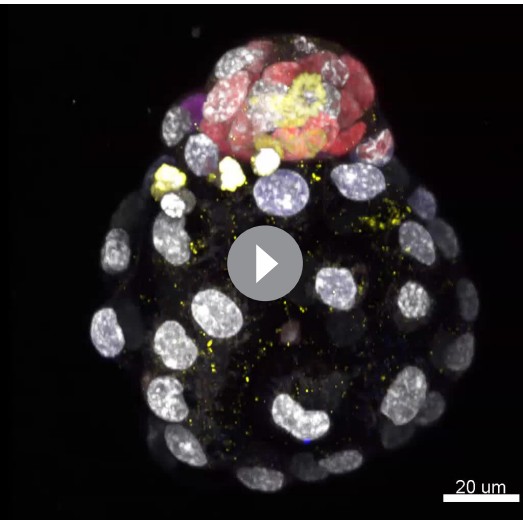

**Video 8.** Fgf4- / - chimera with *Fgf4- / -* ESCs added at morula stage. Yellow surface: ESCs, red nuclei: NANOG, blue nuclei: GATA6, white nuclei: Hoechst.
https://elifesciences.org/articles/56079#video8

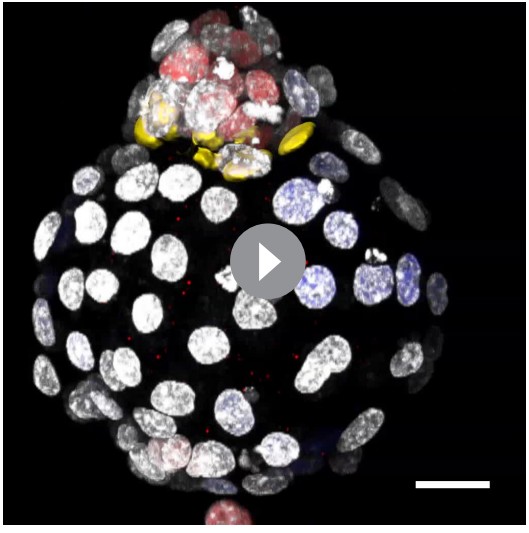

**Video 9.** *Fgf4- / -* chimeras injected with ESCs at the blastocyst stage. Yellow surface: ESCs, red nuclei: NANOG, blue nuclei: GATA6, white nuclei: Hoechst.
https://elifesciences.org/articles/56079#video9

*Tosenberger et al., 2017*) and that the identity of neighbor cells affects fate choice (*Fischer et al., 2020*). In agreement with this view, our proposed configuration of the regulatory network leads to spontaneous divergence of fates among neighboring cells: high availability of growth factor around cells with high NANOG levels (FGF4 producers [*Frankenberg et al., 2011*; *Guo et al., 2010*; *Nowotschin et al., 2019*; *Ohnishi et al., 2014*]) induces NANOG downregulation and PrE fate among surrounding progenitors. Internalization of ligand-receptor complexes by receiving cells, differential intracellular feedback and the transient tandem expression of FGF receptors 1 and 2 in cells adopting a PrE fate (*Kang et al., 2017*; *Molotkov et al., 2017*; *Nowotschin et al., 2019*; *Ohnishi et al., 2014*) may further contribute to the directionality of this local gradient and the resulting induction of opposite fates in neighboring cells.

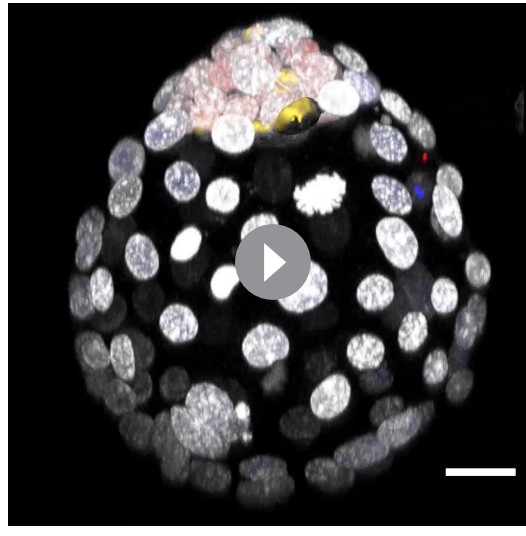

**Video 10.** Fgf4-/-chimerasinjectedwith ESCs at the blastocyst stage. Yellow surface: ESCs, red nuclei: NANOG, blue nuclei: GATA6, white nuclei: Hoechst.
https://elifesciences.org/articles/56079#video10

Further support for the role of FGF4 in the epiblast/PrE decision comes from the finding that introduction of wt ESCs can rescue the all-epiblast ICM found in *Fgf4*$^{-/-}$ embryos (*Figure 7D*). Treatment of *Fgf4*$^{-/-}$ embryos with saturating doses of recombinant FGF4 results in differentiation of all ICM cells toward PrE, instead of a normal balance of epiblast and PrE cells (*Kang et al., 2013*; *Krawchuk et al., 2013*), presumably due to homogeneous availability of high levels of ligand to all progenitor cells within the ICM. In our chimeras, however, the ESCs introduced effectively act as a wt epiblast and provide a local source of FGF4. Consequently, the size of the PrE population is dictated by the amount of ESCs present (*Figure 7C*), suggesting that only progenitors neighboring ESCs are exposed to the signal and adopt a PrE fate, in agreement with in vitro observations (*Raina et al., 2020*). Although our current data lack the spatial resolution to determine the relative position of emerging PrE cells, our experiments establish a direct link between both fates via FGF4 and serve as a proxy for PrE specification in a wt context.

Our study uncovers how cell fate choice and lineage size are coupled via growth factor signaling to ensure robust patterning and morphogenesis in a self-organizing developmental system – independently of size, and without the need for morphogen gradients. These findings provide a framework for our current understanding of signaling and cell fate decisions in the early mammalian embryo and may be generalizable to the formation of other autonomous developmental units.

## Materials and methods

### Mouse strains and husbandry

All animal work was approved by Memorial Sloan Kettering Cancer Center's Institutional Animal Care and Use Committee. Animals were housed in a specific pathogen-free facility under a 12 hr light cycle (6am-6pm). Embryos were obtained from natural matings of 4- to 12 week-old females to stud males. Alleles used and their original source are summarized in *Supplementary file 1*. All mice were maintained on a mixed genetic background.

### Embryo staging

Embryonic days (E) were determined by considering noon of the day of detection of the copulation vaginal plug as E0.5. Higher-resolution staging based on total cell number was used throughout as previously (*Kang et al., 2013*; *Plusa et al., 2008*). To determine the initial developmental stage in experiments involving in vitro culture of blastocysts without a ubiquitous nuclear reporter, a subset

of embryos from each litter (2–4 embryos) were fixed at the time of collection, as previously (*Grabarek et al., 2012*; *Saiz et al., 2016b*). These embryos serve as a reference for the developmental stage of the entire litter (referred to as Littermates in data tables). In all experiments, individual litters were treated as experimental units to reduce variability due to stage differences between litters and to control for batch effects.

## Embryo recovery and handling

Embryos were flushed from dissected oviducts (8–16 cell stage) or uterine horns (blastocyst stages) using forcing Flushing and Holding Media (FHM, Millipore), as previously described (*Behringer et al., 2014*). Live embryos were manipulated in FHM. All solutions used to handle or recover live embryos were pre-warmed at 37˚C. Whenever not being handled under the microscope, embryos were kept at 37˚C in a dry incubator box. Zona pellucidae were removed from embryos by brief washes in Acid Tyrode's solution (Millipore) (*Behringer et al., 2014*) and returned to FHM as soon as the zona dissolved. Blastocysts were fixed in a solution of 4% paraformaldehyde (PFA, Bio-Rad) in phosphate buffered saline (PBS) for 10 min at room temperature and preserved in PBS at 4˚C.

## Embryo genotyping

Embryos were lysed for genotyping in 10 µl of lysis buffer (10 mM Tris, pH 8.5, 50 mM KCl, 0.01% gelatin and 300 µg/ml of Proteinase K) for 50 min at 50˚C, followed by 10 min at 90˚C to inactivate Proteinase K (*Artus et al., 2005*). 2 µl of the lysate was used for genotyping (primer sequences indicated in *Supplementary file 2*).

## Embryo culture and live imaging

Embryos were cultured on 35 mm Petri dishes (Falcon) within microdrops (10–15 µl) of Potassium Simplex Optimized Media with amino acids (KSOM-AA, Millipore) under mineral oil (Sigma), at 37˚C, in a humidified 5% $CO_2$ atmosphere. Prior to culture, embryos were rinsed 3x in drops of KSOM-AA. For live imaging, embryos were cultured on 35 mm glass-bottom dishes (MatTek) within 5–10 µl drops of KSOM-AA under mineral oil.

Images of live embryos were acquired using Zeiss LSM880 laser-scanning confocal microscopes. In cell ablation experiments, images were acquired using a Zeiss C-Apochromat 40x/NA1.1/WD0.21mm objective. For all other experiments, images were acquired using a Zeiss EC Plan-Neofluar 40x/NA1.3/WD0.17mm. GFP was excited using a 488 nm Argon laser at 20µW. mKate2 was excited using a 543 nm HeNe laser or a 561 nm DPSS 561–10 laser at 90µW. Laser power was measured through a Zeiss Plan-Neofluar 10x/NA0.3 objective prior to each imaging session with a light meter (Coherent) and the laser output adjusted to match laser power across experiments. 80 µm stacks were acquired through embryos, at 2 µm intervals, every 15 min. Although these stacks do not capture an entire blastocyst, they encompass the ICM while limiting laser exposure to about 30–40 s per time point and embryo. Time lapse movies were 16–20 hr long.

## Cell lines and culture

Embryonic Stem Cells (ESCs) used were *CAG:H2B-EGFP*[Tg/+] R1 ESCs (*Hadjantonakis and Papaioannou, 2004*) and *Hex*[Venus/+]; *CAG:H2B-tdTomato*[Tg/+] E14 ESCs (*Morgani et al., 2013*). ESCs were cultured feeder-free, on 0.1% gelatin-coated tissue culture dishes (Falcon) in DMEM supplemented with 2 mM L-glutamine, 0.1 mM MEM non-essential amino acids (NEAA), 1 mM sodium pyruvate, 100 U/ml penicillin, 100 µg/ml streptomycin (all from Life Technologies), 0.1 mM 2-mercaptoethanol (Sigma), 10% Fetal Bovine Serum (Hyclone) and 1000 U/ml of recombinant leukemia inhibitory factor (LIF). Cells were passaged by brief incubation at 37˚C in Trypsin-EDTA (0.25% or 0.05%, Life Technologies), neutralization with Serum/LIF media, centrifugation and resuspension in the desired culture media before replating. ESC phenotype was verified visually using fluorescence and cells tested negative for mycoplasma using the MycoAlert PLUS detection kit (Lonza) (N = 2 independent experiments).

## Embryo-embryo aggregation chimeras

Embryos were isolated in the morning of the third day of development (E2.5) and those with four or fewer cells were discarded. Morulae (8–16 cells) from CD1 x *CAG:H2B-EGFP*$^{Tg/+}$ (*Hadjantonakis and Papaioannou, 2004*) crosses were sorted for GFP expression. GFP+ embryos (*CAG:H2B-EGFP*$^{Tg/+}$) were used in the generation of both wt ↔ GFP+ and *Gata6*$^{-/-}$ ↔ GFP+ chimeras (*Figure 1B*), whereas GFP- wild type (wt) morulae were either used for wt ↔ GFP+ chimeras or cultured as un-manipulated controls. For *Gata6*$^{-/-}$ ↔ GFP+ chimeras, the *Gata6*$^{-/-}$ component was generated as in *Figure 1D* (see *Supplementary file 1* for alleles). The use of *Gdf9*$^{iCre}$ (*Lan et al., 2004*) and a floxed *Gata6* (*Sodhi et al., 2006*) gave greater numbers (>25%) of *Gata6*$^{-/-}$ embryos than would result from standard heterozygous *inter se* crosses.

After removal of the zona pellucida, uncompacted, 8cell stage embryos, were disaggregated and blastomeres reaggregated in the desired proportions, as described elsewhere (*Behringer et al., 2014*; *Eakin and Hadjantonakis, 2006*; *Mintz, 1964*; *Tarkowski and Wróblewska, 1967*) and summarized in *Figure 1A,D*. mz*Gata6*$^{-/-}$ embryos are not always null and can show mosaicism. Therefore, in all *Gata6*$^{-/-}$ ↔ GFP+ chimeras the *Gata6*$^{-/-}$ complement was clonal. All *Gata6*$^{-/-}$ ↔ GFP+ chimeras were genotyped retrospectively for the presence of both the conditional and null *Gata6* alleles, and phenotyped based on the presence/absence of GATA6 in immunofluorescence images. Only chimeras with a *Gata6* null GFP- compartment were analyzed.

## ESC-embryo chimeras

ESCs and aggregation dishes were prepared as described in *Behringer et al., 2014*. E2.5 embryos (8–16 cells) of the desired genotype were collected in the morning, denuded and placed within the indentations on the culture dish. Subsequently, ESCs were collected, rinsed twice in KSOM-AA to dilute the FBS and LIF present in the cell media and clumps of the desired number of ESCs placed in contact with each embryo (see *Figure 3—figure supplement 1A*, *Figure 7—figure supplement 1B* for cell numbers). Aggregates were then allowed to develop in vitro for 48–56 hr (*Figure 3A*), until the late blastocyst stage, when they were fixed in 4% PFA.

For blastocyst ESC injections, a PrimeTech Piezo drive (Sutter Instruments) attached to an Eppendorf CellTram microinjectiong system was used to assist with injections. Blastocyst-stage embryos were collected in the morning of the 4th day of development and monitored for development to the mid-blastocyst stage (chimeric embryos were retrospectively estimated to have had ~60–80 cells at the time of injection). Individual embryos were held with a holding pipette (VacuTip, Eppendorf) by the ICM end, while a Piezo Drill flat tip needle (Eppendorf) was used to introduce ESCs into the blastocyst cavity (*Figure 7—figure supplement 1G*). After injection, each litter was allowed to develop for 28–31 hr in KSOM-AA. 3–4 hr after injection, embryos were denuded to allow for cavity expansion while maintaining blastocyst morphology. Reference littermates were fixed in 4% PFA right after injection into the rest of the embryos.

## Laser cell ablation

Ablation experiments are summarized in *Figure 5A and C*. To identify PrE cells, we used a nuclear reporter for *Pdgfra* expression (*Pdgfra*$^{H2B-GFP/+}$ [*Hamilton et al., 2003*; *Plusa et al., 2008*]) (*Figure 5—figure supplement 1A*). To visualize epiblast cells, we combined it with a spectrally distinct, ubiquitous nuclear mKate2 reporter (*ROSA26:CAG:3x-nls-mKate2*$^{Tg/Tg}$, henceforth referred to as *R26:mKate*$^{Tg/Tg}$) (*Susaki et al., 2014*), in which epiblast cells are labeled by nuclear mKate2, but not GFP (*Figure 5—figure supplement 1A,B*). For ablation of PrE cells, most embryos used were obtained from CD1 females intercrossed with *Pdgfra*$^{H2B-GFP/+}$ stud males (*Figure 5A*). For ablation of epiblast cells (and some PrE ablation experiments) all embryos used were from intercrosses of *R26:mKate*$^{Tg/Tg}$ females and *Pdgfra*$^{H2B-GFP/+}$; *R26:mKate*$^{Tg/Tg}$ males (*Figure 5C*).

Embryos were collected at sequential time points between noon and late evening of the fourth day of development (~E3.5 to E4.0) to cover the period of PrE and epiblast specification (~30 to 110 cells, *Figure 5—figure supplement 1E*). To determine the number of PrE or epiblast cells in each embryo (and the total cell number in embryos expressing mKate2), a z-stack through each embryo was acquired prior to cell ablation (*Figure 5A,C*). Nuclei were considered to belong to the PrE if the GFP signal from the *Pdgfra*$^{H2B-GFP}$ allele was markedly higher than in their neighbors on each optical plane (*Figure 5B*; *Figure 5—figure supplement 1A*). Nuclei with intermediate levels of GFP were

scored as putative progenitor cells, whereas nuclei with no GFP (but labeled with mKate2) were scored as epiblast (*Figure 5—figure supplement 1A,B*, see *Figure 5—figure supplement 1D–E* for estimates of each population size in live embryos compared to fixed samples stained for molecular markers). Using time-lapse imaging, we confirmed that cells classified as PrE maintained or upregulated *Pdgfra* expression as they developed, whereas epiblast cells maintained low levels (*Figure 5—figure supplement 1F*, see Data Processing, below, for details on cell classification). On the other hand, cells classified as progenitors upregulated or downregulated *Pdgfra* expression over time, as they adopted PrE or epiblast identity, respectively (*Figure 5—figure supplement 1F*).

Cells were eliminated by repeatedly focusing the beams of an 800 nm Ti:Sapphire femtosecond laser at 10–12% output (Coherent), onto the central region of each nucleus to be ablated (approximately 30–50% of the nuclear area in the section). The number of pulses was determined empirically on a test embryo for each experiment, so that a clear wound was observed on the fluorescent nucleus, but no obvious damage was inflicted to the neighboring cells (typically between 125-250x iterations) (*Figure 5B,D*). Although target nuclei were selected at random throughout the ICM, when possible, nuclei located on the same z plane were targeted, to minimize the length of the procedure. Intact control embryos were only subject to the initial imaging step to estimate the size of each ICM population. Random control embryos were *R26:mKate^{Tg/Tg}* embryos (not carrying the *Pdgfra^{H2B-GFP}* allele) in which a number of ICM cells equivalent to the total number of PrE or epiblast cells found in GFP+ littermates (100% PrE- or epiblast-equivalent) was randomly selected and ablated using the same settings.

After cell ablation, live intact and targeted embryos, were imaged for 16–20 hr, as described above. Laser ablation generated a visible wound in nuclei expressing H2B-GFP, which allowed tracking of targeted and intact cells and assessing nuclear fragmentation as a mark of cell death (*Figure 5E*; *Video 4*). However, mKate2 was extinguished after repeated laser illumination and consequently targeted epiblast cells generally could not be tracked. After time-lapse imaging, embryos were allowed to develop further in an incubator, until a total of 24 hr after the time of collection, before being fixed individually in 4% PFA.

## Immunofluorescence

Whole-mount embryo immunofluorescence was performed as described previously (*Saiz et al., 2016b*; *Saiz et al., 2016a*). Primary antibodies and the dilutions used are provided in *Supplementary file 3*. Secondary antibodies were from Life Technologies, except the AlexaFluor488 anti-chicken, which was from Jackson ImmunoResearch.

## Image acquisition of fixed samples

Immunolabeled embryos were mounted on 35 mm glass-bottom dishes (MatTek), within micro drops of a 5 µg/ml solution of Hoechst 33342 in PBS and imaged using a Zeiss LSM880 laser-scanning confocal microscope, equipped with an oil-immersion Zeiss EC Plan-Neofluar 40x/NA1.3/WD0.17mm. Z-stacks were acquired through whole embryos with 1 µm step between optical slices. Laser power was measured for each laser line prior to each imaging session and parameters adjusted so as to keep laser power consistent for each primary-secondary antibody combination across experiments over time – except for the 405 channel, which was used to excite the nuclear label (Hoechst 33342), and was solely used for image segmentation.

## Image processing

Nuclear image segmentation of still images and manual image correction was performed using the MINS software (*Lou et al., 2014*) as previously described (*Morgani et al., 2018b*; *Saiz et al., 2016a*). MINS is freely available at https://github.com/therealkatlab/MINS (requires MATLAB license). Missing nuclei, or multiple nuclei segmented as one (under-segmentation) were measured manually using ImageJ (Rasband, W.S., ImageJ, U. S. National Institutes of Health, Bethesda, Maryland, USA, https://imagej.nih.gov/ij/, 1997–2018) and the measured values for each channel, as well as XYZ coordinates, added to the data table.

Time lapse images were processed using Imaris (Oxford Instruments). Individual ICM cells were identified based on the presence of H2B-GFP and/or nuclear mKate2 and tracked manually over the course of the movie using the spots function. Cell death or mitotic events were labeled as such for

each individual cell (*Figure 5—figure supplement 1G*). GFP levels are a proxy for *Pdgfra* expression and allow the classification of cell types over time (*Figure 5—figure supplement 1F,G*). Cell identity was assigned visually based on GFP levels at the time of the experiment (*Figure 5—figure supplement 1A,E*) and verified retrospectively (and reclassified when necessary) after quantification of GFP levels.

## Data processing

Fluorescence data obtained after segmentation with MINS and manual curation was processed as described in *Morgani et al., 2018b*; *Saiz et al., 2016b*, with certain modifications for subsets of the data. For samples where GFP fluorescence was detected indirectly with an anti-GFP antibody (*Figure 1*; *Figure 1—figure supplement 2*), anti-GFP::AF488 fluorescence was log-transformed and corrected for fluorescence decay along the Z axis by fitting a linear model, as described in *Saiz et al., 2016a*. For all other samples (live or fixed), an Empirical Bayesian slope correction step was applied to log-transformed, Z-corrected data, as in *Saiz et al., 2016b*. Detailed descriptions of the correction steps can be found in https://github.com/nestorsaiz/saiz-et-al_2020/blob/master/notebooks/_data-processing-wf.pdf and in https://github.com/nestorsaiz/saiz-et-al_2020/blob/master/notebooks/Z-correction.ipynb (*Saiz et al., 2020*).

Two different antibodies were used to detect NANOG and GATA6 expression (*Supplementary file 3*). Rabbit anti-NANOG (NANOG(rb)) and goat anti-GATA6 (GATA6(gt)) were used for most samples, as previously (*Kang et al., 2017*; *Morgani et al., 2018b*; *Saiz et al., 2016b*; *Schrode et al., 2014*). When a rat anti-NANOG (NANOG(rat)) or a rabbit anti-GATA6 (GATA6(rb)) were used (*Supplementary file 3*), fluorescence values were transformed to NANOG(rb)- and GATA6(gt)-equivalents, respectively, using a linear regression model generated from samples stained with two antibodies for each marker. A more detailed description can be found in https://github.com/nestorsaiz/saiz-et-al_2020/blob/master/notebooks/nanogata-tx.ipynb (*Saiz et al., 2020*).

Cell identity was assigned to ICM cells based on relative NANOG and GATA6 expression, rescaled against their maxima in each litter to account for variability between litters and over time. When separated based on NANOG and GATA6 levels, ICM cells typically form four clusters, corresponding to the epiblast (NANOG+), PrE (GATA6+), double positive (DP, progenitor cells) and NANOG-low or NANOG- epiblast (NANOG-lo) (*Kang et al., 2017*; *Morgani et al., 2018b*; see *Saiz et al., 2016b*). A fifth category are double negative (DN) cells, which do not express NANOG or GATA6. These are rare and often correspond to mitotic, dying cells, or uncorrected segmentation errors. To automatically identify cell populations in the ICM we implemented hierarchical clustering. We determined empirically that the agglomerative UPGMA (unweighted pair method with arithmetic mean) captured the PrE and epiblast clusters comparably to k-means clustering (as used in *Kang et al., 2017*; *Morgani et al., 2018b*; *Saiz et al., 2016b*) across all blastocyst stages, however, it outperformed k-means when classifying DP and NANOG-low epiblast cells. More detailed descriptions of the transformation and classification steps can be found in https://github.com/nestorsaiz/saiz-et-al_2020/blob/master/notebooks/H-clustering.ipynb and in https://github.com/nestorsaiz/saiz-et-al_2020/blob/master/notebooks/GFP-classification.ipynb (*Saiz et al., 2020*).

Corrected fluorescence values of H2B-GFP (*Pdgfra* expression) for ICM cells were used to determine changes in cell identity over time in time-lapse movies. To reduce noise in the data, a moving average was calculated for each cell, with a window of 4 timeframes (1 hr). A simple classifier was thus devised to assign cell identity automatically to individual cells based on these GFP levels. Thresholds for fluorescence were manually determined for each litter analyzed, both at the start and the end of the movie, which were used to determine a threshold slope for PrE and epiblast identities. The classifier followed two rules: (1) cells classified as PrE or epiblast maintain that identity for the remainder of the movie – based on *Saiz et al., 2016b*; *Xenopoulos et al., 2015* and the observed lack of oscillations on GFP levels or obvious, systematic shifts from high to low levels, or vice-versa (*Figure 5—figure supplement 1F,G*) – (2) cells classified as DP at t = 0 become PrE or epiblast once their level of GFP remains above or below the respective thresholds for at least 2 hr – to account for fluctuations and noise in data (*Figure 5—figure supplement 1G*).

## Acknowledgements

We thank Drs Frederic Geissmann, Pierre-Louis Loyher, Alison North and Christina Pyrgaki for access to their two-photon systems and for invaluable help in setting up the experimental conditions for cell ablation in embryos; Dr Pavak Shah for advice on laser ablation in different contexts; Drs Jennifer Nichols and Stanley Strawbridge for discussions of complementary projects; Drs Alfonso Martinez Arias, Berenika Plusa, Pedro Rocha, Christian Schröter, Eric Siggia, Philippe Soriano, Joana Vidigal, and members of the Hadjantonakis lab for insightful discussions and critical feedback. The authors dedicate this work to the memory of Prof Andrzej K Tarkowski and Dr Yoshiki Sasai, whose pioneering work has long been a source of inspiration.

## Additional information

### Funding

| Funder | Grant reference number | Author |
|---|---|---|
| Eunice Kennedy Shriver National Institute of Child Health and Human Development | R01-HD094868 | Anna-Katerina Hadjantonakis |
| National Institute of Diabetes and Digestive and Kidney Diseases | R01-DK084391 | Anna-Katerina Hadjantonakis |
| National Cancer Institute | P30-CA008748 | Anna-Katerina Hadjantonakis |
| Spanish Ministry of Science and Innovation | PGC2018-101251-B-I00 | Jordi Garcia-Ojalvo |
| Spanish Ministry of Science and Innovation | CEX2018-000792-M | Jordi Garcia-Ojalvo |
| Catalan Institution for Research and Advanced Studies | | Jordi Garcia-Ojalvo |
| Starr Foundation | | Néstor Saiz |

The funders had no role in study design, data collection and interpretation, or the decision to submit the work for publication.

### Author contributions

Néstor Saiz, Conceptualization, Data curation, Formal analysis, Funding acquisition, Investigation, Visualization, Methodology, Writing - original draft, Writing - review and editing; Laura Mora-Bitria, Formal analysis, Investigation, Methodology; Shahadat Rahman, Hannah George, Jeremy P Herder, Data curation; Jordi Garcia-Ojalvo, Software, Formal analysis, Supervision, Funding acquisition, Investigation, Visualization, Writing - original draft, Project administration, Writing - review and editing; Anna-Katerina Hadjantonakis, Conceptualization, Resources, Supervision, Funding acquisition, Visualization, Writing - original draft, Project administration, Writing - review and editing

### Author ORCIDs

Néstor Saiz https://orcid.org/0000-0003-0637-791X
Shahadat Rahman http://orcid.org/0000-0002-3424-6768
Jordi Garcia-Ojalvo https://orcid.org/0000-0002-3716-7520
Anna-Katerina Hadjantonakis https://orcid.org/0000-0002-7580-5124

### Ethics

Animal experimentation: All animal work was approved by Memorial Sloan Kettering Cancer Center's Institutional Animal Care and Use Committee (Protocol 03-12-017, Hadjantonakis PI).

### Decision letter and Author response

Decision letter https://doi.org/10.7554/eLife.56079.sa1

Author response https://doi.org/10.7554/eLife.56079.sa2

## Additional files

### Supplementary files

- Supplementary file 1. List of alleles used in the study.
- Supplementary file 2. List of primers used for genotyping.
- Supplementary file 3. List of antibodies used for immunofluorescence.
- Transparent reporting form

### Data availability

All image data processing was done in R version 3.4.2, using RStudio as an interactive development environment. All processed data as well as the code used to transform data and classify cells is available at https://github.com/nestorsaiz/saiz-et-al_2020 (copy archived at https://github.com/elifesciences-publications/saiz-et-al_2020). All raw confocal images and data tables will be freely available on Figshare with DOI 10.6084/m9.figshare.c.4736507. Code for phase-plane analysis and modeling is available at https://github.com/jgojalvo/EmbryoRobustness (copy archived at https://github.com/elifesciences-publications/EmbryoRobustness).

The following dataset was generated:

| Author(s) | Year | Dataset title | Dataset URL | Database and Identifier |
|---|---|---|---|---|
| Saiz N, Mora-Bitria L, Rahman S, George H, Herder J, Garcia-Ojalvo J, Hadjantonakis AK | 2020 | Growth factor-mediated coupling between lineage size and cell fate choice underlies robustness of mammalian development | https://doi.org/10.6084/m9.figshare.c.4736507 | figshare, 10.6084/m9.figshare.c.4736507 |

The following previously published datasets were used:

| Author(s) | Year | Dataset title | Dataset URL | Database and Identifier |
|---|---|---|---|---|
| Saiz N, Williams MK, Seshan VE, Hadjantonakis AK | 2016 | Asynchronous fate decisions by single cells collectively ensure consistent lineage composition in the mouse blastocyst. | https://doi.org/10.6084/m9.figshare.c.3447537.v1 | figshare, 10.6084/m9.figshare.c.3447537.v1 |
| Morgani SM, Saiz N, Garg V, Raina D, Simon CS, Kang M, Arias AM, Nichols J, Schroter C, Hadjantonakis AK | 2018 | A Sprouty4 reporter to monitor FGF/ERK signaling activity in ESCs and mice. | https://doi.org/10.6084/m9.figshare.c.4142081 | figshare, 10.6084/m9.figshare.c.4142081 |

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

## Appendix 1

### Growth-factor-mediated coupling between lineage size and cell fate choice underlies the robustness of mammalian development

#### A1.1 Molecular model

The minimal model used in this paper can be derived from the following kinetic model describing the interactions between Nanog (N), Gata6 (G) and FGF (F) shown in *Appendix 1—figure 1*:

$$\frac{dN}{dt} = \frac{\alpha_n}{1 + (\langle F \rangle G / K_g)^m} - \delta_n N \tag{A1}$$

$$\frac{dG}{dt} = \frac{\alpha_g \langle F \rangle}{1 + (N / K_n)^n} - \delta_g G \tag{A2}$$

$$\frac{dF}{dt} = \alpha_f N - \delta_f F \tag{A3}$$

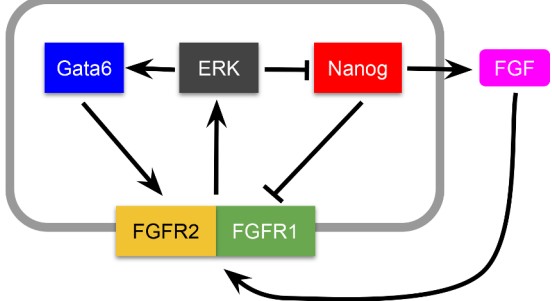

**Appendix 1—figure 1.** Potential molecular mechanism underlying the Epi-PrE decision. We note, nevertheless, that the minimal model obtained below could also represent other molecular architectures of the Nanog-Gata6 circuit, and thus can be interpreted as a general representation of a growth-factor-mediated mutual inhibition.

Here Nanog and Gata6 in a given cell inhibit each other indirectly, through FGF signaling coming from the cell itself and from its neighbors (represented by the average term $\langle F \rangle$ in *Equations A1 and A2*). The average FGF term, $\langle F \rangle$, enters the two inhibition terms asymmetrically. First, since Nanog is inhibited by ERK after FGF activates its receptor (which is itself activated by Gata6), we consider that the input function of Nanog (first term in the right-hand side of *Equation A1*) is an inhibitory Hill function in which FGF and Gata6 act together in the denominator. On the other hand, since Nanog is assumed to inhibit directly the receptor, the input function of Gata6 (first term in the right-hand side of *Equation A2*) is an inhibitory Hill function in which Nanog is the only inhibitory element, while FGF acts as a pre-factor representing the signaling.

#### A1.2 Reduction to one variable

We now assume that Gata6 and FGF are faster than Nanog, so that *Equations (A2) and (A3)* can be considered to be in quasi-steady state. We thus can make the right-hand side of those equations equal to zero, and substitute the resulting expressions of G and F into *Equation (A3)*. Nondimensionalization and a minor algebraic rearrangement of the production term lead to a single ODE per cell:

$$\frac{dx_i}{dt} = \frac{\alpha (1 + x_i^n)^m}{(1 + x_i^n)^m + (\langle x \rangle_i / K)^{2m}} - x_i \tag{A4}$$

where $x_i$ represents the concentration of Nanog in cell i measured in units of $K_n$ and time is measured in units of $1/\delta_n$. The signaling term $\langle x \rangle_i$ denotes the average value of x in the neighborhood of cell i, including $x_i$ itself, with weights that can be fixed at will. Here we assume equal weights for cells

closer than a certain coupling range (given by the factor $f_{\text{range}}$, see next section), and 0 weights for the rest of the cells. The dimensionless parameters $\alpha$ and K are related with the original parameters of model *Equations (A1–A3)* through the following relationships:

$$\alpha \equiv \frac{\alpha_n}{K_n \delta_n}, \qquad K \equiv \frac{\delta_f}{\alpha_f K_n} \sqrt{\frac{\delta_g K_g}{\alpha_g}} \tag{A5}$$

The dimensionless parameters of the model that we use in this paper, together with those of the agent-based model described in what follows, are given in *Appendix 1—table 1* below.

**Appendix 1—table 1.** Parameter values.
The parameters of the biochemical circuit (top four parameters, above the line) are given in dimensionless units. The parameters of the agent-based model (below the line) are given in arbitrary units, with time being rescaled a posteriori to match the experimental observations approximately.

| Parameter | Description | Value |
|---|---|---|
| $\alpha$ | maximum expression strength | 10 |
| K | inhibition threshold | 0.9 |
| N | Nanog inhibition cooperativity | 2 |
| m | Gata6 inhibition cooperativity | 2 |
| $f_{\text{range}}$ | FGF coupling range factor | 1.2 |
| $m_i$ | initial mass | $10^{-6}$ |
| $r_i$ | initial radius | 5 |
| b | effective friction coefficient | $10^{-6}$ |
| $K_0$ | adhesion strength | $10^{-4}$ |
| $f_{\text{adh}}$ | adhesion strength reduction for different fates | 1.5 |
| $\mu$ | adhesion range | 2 |
| $\tau_{\text{div}}$ | average division time | 10 |
| $\sigma_{\text{div}}$ | relative division time noise | 0.5 |
| $x_0$ | initial x | 3.0 |
| $\sigma_x$ | relative partition noise | 0.01 |
| $N_{\text{circ}}$ | circuit turn-on average ICM size | 20 |
| $\sigma_{\text{N,circ}}$ | circuit turn-on relative noise | 0.1 |
| $f_{\text{min}}$ | circuit turn-off lower factor | 0.05 |
| $f_{\text{max}}$ | circuit turn-off upper factor | 0.95 |
| $f_{\text{PrE}}$ | PrE fate factor | 0.2 |
| $f_{\text{EPI}}$ | EPI fate factor | 0.8 |
| $\tau_{\text{circ}}$ | circuit turn-on average time (chimera sims.) | 45.0 |
| $\sigma_{\tau,\text{circ}}$ | circuit turn-on relative noise (chimera sims.) | 0.02 |
| $r_{\text{ESC}}$ | ESC radius (chimera sims.) | 2.0 |

## A1.3 Agent-based modeling

We apply the biochemical model described above to a population of proliferating spherical cells that simulate the developing embryo (*Tosenberger et al., 2017*), whose mechanics are simply governed by:

$$m_i \frac{d^2 \vec{x}_i}{dt^2} = -b \frac{d\vec{x}_i}{dt} + \sum_{j=1}^{N} \vec{F}_{ij}, \qquad i = 1, 2 \ldots, N \tag{A6}$$

Here $\vec{x}_i$ denotes the position of the center of cell $i$ with mass $m_i$, and $\vec{F}_{ij}$ represents the force

between cells $i$ and $j$, which is repulsive for separations between cell centers smaller than the sum of their radii, $r_{ij} \equiv r_i + r_j$, attracting for larger separations up to $\mu r_{ij}$, and zero for separations larger than $\mu r_{ij}$. This is represented by:

$$\vec{F}_{ij} = \begin{cases} F_0 \left(\frac{r_{ij}}{d} - 1\right)\left(\frac{\mu r_{ij}}{d} - 1\right)\frac{1}{d}(\vec{x}_i - \vec{x}_j) & \text{if } d < \mu r_{ij} \\ 0 & \text{if } d > \mu r_{ij} \end{cases} \tag{A7}$$

where $d$ is the Euclidean distance between the cell centers. The interaction strength $K_0$ between two cells is made to depend on their relative fates, being smaller between cells of different type than between cells of the same type (see *Appendix 1—table 1*). In the biochemical circuit described above, *Equation (A4)*, two cells are considered neighbors (i.e. they can detect their mutual FGF levels) if the separation between their centers is smaller than the sum of their radii multiplied by a factor $f_{\text{range}}$.

Cellular proliferation is modeled by making each cell divide a certain time after its previous division. We consider that this time varies uniformly in the interval $[\tau_{\text{div}}(1 - \sigma_{\text{div}}), \; \tau_{\text{div}}(1 + \sigma_{\text{div}})]$. The daughter cells are placed at symmetrically selected positions around the center of the mother cell along a randomly chosen direction, separated a distance equal to the radius of the mother cell. Upon each division, cells (with the exception of externally added embryonic stem cells, see below) reduce their mass a factor two and their radius a factor $\sqrt[3]{2}$, which corresponds to their volume being halved. The daugther cells inherit all other properties from their mother, including their velocity, mass, fate, and Nanog level $x$. Nanog is partitioned stochastically upon division, represented by a uniformly distributed random amount with an amplitude $\sigma_x$ relative to its average level.

The biochemical circuit is turned on after the ICM has reached a certain average ICM size $N_{\text{circ}}$ (with relative noise $\sigma_{\text{N,circ}}$), to account for the fact that experimental observations do not reveal any meaningful dynamics of Nanog nor Gata6 in the first stages of embryonic development. The scaling experiments are modeled by multiplying $N_{\text{circ}}$ by the corresponding factor (2x, 0.5x, 0.25x). In the chimera simulations, circuit turn-on is applied instead at a specific average time $\tau_{\text{circ}}$ (with relative noise $\sigma_{\tau,\text{circ}}$) that is consistent with the turn-on ICM size exhibited by the wild-type embryos.

The fate of a cell is determined on the basis of the value of the $x$ variable in that cell at any given time. Specifically, if $x$ is larger than a fraction $f_{\text{EPI}}$ of its maximum possible value ($x_{\text{M}} = \alpha/(1 + 1/(2K)^{2m})$), the cell is considered to be an EPI cell. In turn, if $x < f_{\text{PrE}} x_{\text{M}}$ the cell is considered a PrE cell. These conditions are chosen to mimic the way in which cell fates are usually assigned experimentally (*Saiz et al., 2016b*). Once those fates are reached, the biochemical circuit is turned off. Specifically, $x(t)$ is only updated following the dynamics described by *Equation (A4)* when $f_{\text{min}} x_{\text{M}} < x < f_{\text{max}} x_{\text{M}}$. This assumption reflects the fact that our biochemical models ignores downstream processes that follow EPI and PrE specification, making these cell-fate choices irreversible (*Xenopoulos et al., 2015*). The limit factors $f_{\text{PrE}}$ and $f_{\text{min}}$ (resp. $f_{\text{EPI}}$ and $f_{\text{max}}$) are assumed to differ somewhat (see *Appendix 1—table 1*), in order to make the irreversible character of the decision robust to noise in $x$.

The ablation experiments are modeled by eliminating random sets of cells (within a specific fate if applicable) at specified ICM sizes, and allowing the remaining cells to redistribute according to the dynamics of *Equation (A6)*. The chimera experiments are modeled by adding cells with a distinct (ESC) fate, which do not obey the biochemical dynamics described by *Equation (A4)* and have a fixed radius $r_{\text{ESC}}$ that does not decrease upon proliferation. The ESCs are added recursively at a location adjacent to the ICM cell whose position is farthest from the center-of-mass of the simulated embryo at that particular time instant, outside of the embryo and separated from it a distance equal to the radius of that ICM cell. As in the case of the ablation, the added ESCs reorganize quickly after addition, following the dynamics of *Equation (A6)*.

The model parameters used throughout the paper are given in *Appendix 1—table 1*. Note that the cell-fate decision circuit depends only on two dimensionless parameters, $\alpha$ and $K$, plus two Hill coefficients $N$ and $m$. The agent-based model provides the substrate on which the circuit operates, effectively mixing the cells as they proliferate, and providing them with a neighborhood that represents the entire population. The choice of parameters for the agent-based simulations thus does not affect the qualitative behavior of the biochemical circuit introduced here.

