## [Decision Letter]

**Acceptance summary:**

In this study Saiz et al. explore the mechanism responsible for the balanced generation of epiplast and primitive endoderm cells from the inner cell mass of the mouse embryo. The authors use a combination of molecular and embryological assays coupled with mathematical modelling to conclude that non-autonomous regulative feedback, mediated by FGF4, is responsible for robustly maintaining Epi and PrE ratios.

**Decision letter after peer review:**

Thank you for submitting your article "Growth factor-mediated coupling between lineage size and cell fate choice underlies robustness of mammalian development" for consideration by *eLife*. Your article has been reviewed by three peer reviewers, and the evaluation has been overseen by a Reviewing Editor and Naama Barkai as the Senior Editor The following individual involved in review of your submission has agreed to reveal their identity: Jean-Léon Maître (Reviewer #3).

The reviewers have discussed the reviews with one another and the Reviewing Editor has drafted this decision to help you prepare a revised submission.

Summary:

In this study Saiz et al. explore the mechanism responsible for the balanced generation of epiplast and primitive endoderm cells from the inner cell mass of the mouse embryo. The authors use a combination of molecular and embryological assays coupled with mathematical modelling to conclude that non-autonomous regulative feedback, mediated by FGF4, is responsible for robustly maintaining Epi and PrE ratios.

Essential revisions:

The reviewers think this is a very well done study, which can be published without any further experiments. However they all commented on the extend of the modelling presented and suggest that this could be strengthened in 3 specific areas with regard to:

– FGF spread;

– How geometry affects the behaviour;

– How asynchrony in decisions is implemented in the model and relates to the biology.

---

## [Author Response]

Essential revisions:The reviewers think this is a very well done study, which can be published without any further experiments. However they all commented on the extend of the modelling presented and suggest that this could be strengthened in 3 specific areas with regard to:– FGF spread;– How geometry affects the behaviour;– How asynchrony in decisions is implemented in the model and relates to the biology.

Our mathematical model was designed to capture the basic interactions posited to underlie the Epi-PrE decision taking place in the early mouse embryo. In spite of its simplicity, the model allows us to examine the influence of a variety of biological factors, in particular those highlighted by the reviewers: (i) the signaling range of FGF, (ii) the geometry of the population, and (iii) the role of cell division variability on the asynchronicity of cellular decisions.

We have systematically studied the influence of these factors in the behavior of our model, by performing a sensitivity analysis of the final cell fate ratio exhibited by the model, when all model parameters are varied independently by increasing or decreasing their values 20%, 30% and 50% beyond the nominal values given in Appendix 1—table 1. The results are shown in the new Figure 2—figure supplement 1F. We describe these results in what follows.

i) FGF signaling range (spread)

An increase in the range of signaling (right-most set of bars in all three panels of Figure 2—figure supplement 1F) does not noticeably change the decision pattern exhibited by the model, which reproduces qualitatively the experimental observations. Only when the signaling range is substantially reduced (-30% and -50%, middle and right panels in Figure 2—figure supplement 1F) is the decision prevented. This is to be expected, since the latter case a cell only detects the FGF molecules that it itself secretes. In that limit the decision cannot occur, since in our model the decision is fully mediated by intercellular FGF signaling.

We have also systematically varied the range of FGF signaling, from the autocrine limit (left-most bar in Figure 2—figure supplement 1G), in which cells only sense the FGF they produce, to the all-to-all coupling limit (right-most set of bars in Figure 2—figure supplement 1G), in which the FGF secreted by a cell is sensed by all cells in the embryo. The results show that as soon as cells are coupled with their nearest neighbors (second set of bars from the left in Figure 2—figure supplement 1G), the decision is reached with appropriate cell-fate ratios, which are maintained even when signaling extends over the entire population. Local nearest-neighbor coupling is thus not necessary for the correct decision to arise.

ii) Geometry

The decision is also robust to most perturbations in the mechanical parameters of the model, which partly define, and are defined by, the geometry of the system. This can be seen again in Figure 2—figure supplement 1F, which shows that changes in the effective friction and adhesion between cells, and in the adhesion range, do not for the most part affect the cell fate decision ratios produced. Only when the effective friction decreases substantially (right panel in Figure 2—figure supplement 1F) is the decision lost. In this latter case, cell mobility becomes too high and intercellular signaling stops taking place efficiently, which mimics the autocrine limit discussed in the previous point.

A stronger geometrical constraint is the dimensionality of the system. The main results reported in this manuscript were obtained in a three-dimensional configuration, which corresponds to the experimental situation studied here. However, we have also examined how our model behaves in a more constrained two-dimensional topology. The response of the system when cells are restricted to move (and signal) on a 2D plane are displayed in Figure 2—figure supplement 1E. Simulations show that the cell fate decision process is maintained in these conditions, which manifests the robustness of the mechanism to the dimensionality of the system.

iii) Variability and asynchrony of the decision

Given the key role of cell-cell signaling set forth in this study, it is important for our model to describe cell proliferation adequately: when cells divide the population is reorganized, and thereby the signals sensed by cells from their neighbors change. Experimental observations from ourselves and others show that, as the early embryo develops, cells divide in an asynchronous manner.

Following previous modeling approaches (see e.g. (Tosenberger et al., 2017)), we implemented this asynchrony in our model, by considering that the cell division time of each cell varies randomly (according to a uniform distribution) with respect to a baseline in which cells would divide at times proportional to an average division time τdiv. We quantify the variability in terms of the standard deviation of this random variable, which we call the relative division time noise σdiv.

Both τdiv and σdiv were approximated based on our own experimental observations and data published in (Bischoff et al., 2008) and (Plusa et al., 2008). Our sensitivity analysis (Figure 2—figure supplement 1F and H) shows again that the decision process is also very much robust to changes in the asynchrony of cell division. Changes ranging from 0 to 100% variability with respect to the average division time failed to substantially change the resulting proportion of cell types. Noticeably, even in the absence of variability in division times (synchronous divisions, left-most set of bars in Figure 2—figure supplement 1H) the decision holds with appropriate cell fate ratios. This is to be expected given the existence of other sources of variability in the proliferation process, in particular noise in the segregation of NANOG molecules between daughters upon division (parameter named simply partition noise in Figure 2—figure supplement 1F, for which the division is also seen to be robust), and randomness in the 3D orientation of the axis along which cells divide.

In the revised manuscript we discuss the results described above, which we have included in new panels (E-H) that have been added to Figure 2—figure supplement 1. We have also taken the opportunity to revise the text in order to discuss the similarities and differences between our model and previous pattern formation frameworks, such as the Turing and lateral inhibition mechanisms. While our model has conceptual similarities to both scenarios, its specific form, and the way in which it derives from a minimal set of assumptions regarding the regulation of the Epi-PrE cell fate choice, is to our knowledge, new. We have also attempted to clarify further the way in which FGF signaling affects NANOG and GATA6 in our model.